# Changing travel patterns in China during the early stages of the COVID-19 pandemic

Hamish Gibbs[1,2,3 ✉], Yang Liu [1,2,3 ✉], Carl A. B. Pearson [1,2], Christopher I. Jarvis[1,2], Chris Grundy[1], Billy J. Quilty[1,2], Charlie Diamond[1,2], LSHTM CMMID COVID-19 working group* & Rosalind M. Eggo [1,2]

Understanding changes in human mobility in the early stages of the COVID-19 pandemic is crucial for assessing the impacts of travel restrictions designed to reduce disease spread. Here, relying on data from mainland China, we investigate the spatio-temporal characteristics of human mobility between 1st January and 1st March 2020, and discuss their public health implications. An outbound travel surge from Wuhan before travel restrictions were implemented was also observed across China due to the Lunar New Year, indicating that holiday travel may have played a larger role in mobility changes compared to impending travel restrictions. Holiday travel also shifted healthcare pressure related to COVID-19 towards locations with lower healthcare capacity. Network analyses showed no sign of major changes in the transportation network after Lunar New Year. Changes observed were temporary and did not lead to structural reorganisation of the transportation network during the study period.

---

[1] Department of Infectious Disease Epidemiology, London School of Hygiene and Tropical Medicine, Keppel Street, London WC1E 7HT, UK. [2] Centre for Mathematical Modelling of Infectious Diseases, London School of Hygiene and Tropical Medicine, Keppel Street, London WC1E 7HT, UK. [3] These authors contributed equally: Hamish Gibbs, Yang Liu. *A list of authors and affiliations appears at the end of the paper. ✉email: Hamish.Gibbs@lshtm.ac.uk; Yang.Liu@lshtm.ac.uk

The COVID-19 pandemic was first identified in Wuhan, China, in late 2019, and came to prominence in January 2020, and quickly spread within the country. January is also a major holiday period in China, and the 40-day period around Lunar New Year (LNY), or *Chunyun*, marks the largest annual human movement in the world, with major travel flows out of large cities[1]. The purpose of this holiday travel is often to visit family members. The temporary displacement from residential addresses as a result of this holiday travel could last one to two weeks, up to a month. In 2019, nearly 3 billion individual journeys were made during *Chunyun*[2]. In 2020, *Chunyun* lasted from 10th January to 18th February[3], with the first day of the LNY holidays on 24th January, followed by the first day of LNY on 25th January. This period coincided with the initial phase of the COVID-19 pandemic, and there has been speculation that holiday travel may have accelerated the propagation of COVID-19 both within China and internationally[4].

As part of initial efforts to contain the outbreak, the Chinese government announced a cordon sanitaire for the city of Wuhan, Hubei Province, starting on 23rd January 2020, one day before LNY holidays. This intervention restricted all non-essential movement into and out of the city. Services at airports, train stations, long-distance bus stations, and commercial ports were all suspended[5]. Several studies have focused on assessing the effectiveness of the cordon sanitaire in Wuhan and other domestic travel restrictions in China in the context of COVID-19 control[6–8]. As other affected regions worldwide begin implementing similar travel restrictions[9], it is critical to understand human mobility patterns during the initial phase of the COVID-19 pandemic and their potential implications for other countries.

Out-going traffic from Wuhan was reduced by 89% within two days of the *cordon sanitaire*, according to data from Baidu Huiyan, an internet service company in China which uses location targeting to provide services to users. Baidu's Location Based Service (LBS)[10] provides travel fluxes between prefectures in China during the annual *Chunyun* period to allow monitoring of movement of people using their services.

Previous analyses of Baidu movement data have used mobility data in transmission models[6,11], and others have examined the changes in patterns around Wuhan[7]. A key unknown is to what extent the observed travel patterns in Wuhan and the rest of China were part of regular seasonal movements or were responses to the emerging epidemic or interventions against it, including the *cordon sanitaire*. Relying on a range of data scientific techniques, we examine human movement between Chinese prefectures on multiple geographic scales to provide a detailed examination of travel patterns during the early stages of the COVID-19 pandemic in China. We combine analyses of travel patterns from Wuhan, where the first COVID-19 epidemic was identified, and the first Chinese city to introduce large scale movement restrictions, with an analysis of the effects on the overall Chinese travel network. We further explore the relationship between travel patterns during the LNY holidays and regional healthcare capacity, to understand the impact of the human movements on the healthcare pressure caused by the spreading epidemic. This research is intended to provide a complete picture of the overall movement dynamics in China, and the public health implications of those movements, and has relevance to other countries implementing travel restrictions in an effort to limit the spread of COVID-19.

## Results

**Human movement surrounding Wuhan, Hubei**. We used daily prefecture-level movement data across China provided by Baidu Huiyan[10] to understand the spatial and temporal characteristics of movement patterns before, during and after the COVID-19 epidemic in Wuhan. Before the cordon sanitaire and during the initial phase of the COVID-19 epidemic, outbound travel volume from Wuhan was marked by an early-January peak, followed by a sharper second peak in the days before the LNY holidays (Fig. 1a). The first peak was not observed in 2019, while the second peak was higher in 2020 than 2019. Because the start of Wuhan's cordon sanitaire and the beginning of LNY holidays were only one day apart, we refer only to LNY while describing our results.

Using k-means clustering of the timeseries of daily outbound travel from Wuhan to other prefectures, we identified four general temporal patterns that captured the travel patterns from Wuhan (Fig. 1e, Supplementary Table 1). Two of these clusters exhibited an increase in flow immediately before LNY (clusters A and B). Members of clusters A and B are geographically closer to Wuhan (Fig. 1d), with fewer residents and overall lower population density (Fig. 1e, Supplementary Tables 2–3). Cluster C exhibited two peaks around 7 and 22 January 2020, respectively. Cluster D showed one peak in early-January 2020, with no peak immediately preceding the LNY holidays. The findings are not sensitive to the number of clusters, (Supplementary Figs. 1–5).

The earliest detection of COVID-19 outside of Wuhan was 17th January 2020 (Fig. 1b). By late March, over 90% of prefectures and province-level cities (further detail on administrative levels included in "Methods" section) in mainland China had at least one confirmed case of COVID-19. Most prefectures confirmed their first COVID-19 cases between 23rd and 26th January 2020.

Among the four clusters identified, cluster membership was associated with COVID-19 detection timing (p-value = 0.0004). Members of cluster D tended to have earlier COVID-19 detection. Such association persisted after adjusting for surveillance bias (p-value = 0.00002, see also Supplementary Fig. 6). Cluster membership was also associated with differences in prefecture-level population sizes (Fig. 1c). Cluster D includes large population centres (e.g., Beijing, Shanghai, Guangzhou and Shenzhen) (Fig. 1d). After the possible arrival of infected individuals from Wuhan, these highly connected cities could have contributed to the further spread of COVID-19 to places less directly connected to Wuhan. There were also a small number of prefectures that did not have any confirmed cases until 3 weeks after the cordon sanitaire in Wuhan.

We repeated the same analyses for other large cities in China, finding that despite the different numbers of clusters identified, the general patterns in movement flows observed in Wuhan were seen elsewhere in mainland China, with an early January peak in travel, and another increase in travel volume preceding LNY (Supplementary Figs. 6–10). The association between the population size of destinations and geographic distance, however, was less apparent. The early-January peak in Wuhan coincided with the beginning of winter break for university students in China[12], approximately one million of whom study in Wuhan[13]. Without information about the age composition of travellers at this time, we cannot provide a definite explanation of this observation.

There is anecdotal evidence implying an association between the announcement of a cordon sanitaire on 23rd January and temporarily increased outbound travel from Wuhan[14]. This relationship, if true, could have hindered the effectiveness of the *cordon sanitaire*. Focusing on the six-day period preceding LNY, we compared the outbound travel patterns from Wuhan with the rest of mainland China using 2019 as the baseline. We used two variability metrics to investigate potential outbound travel surges: (1) a proportion-based matric, Eq. (3), that captures the relative

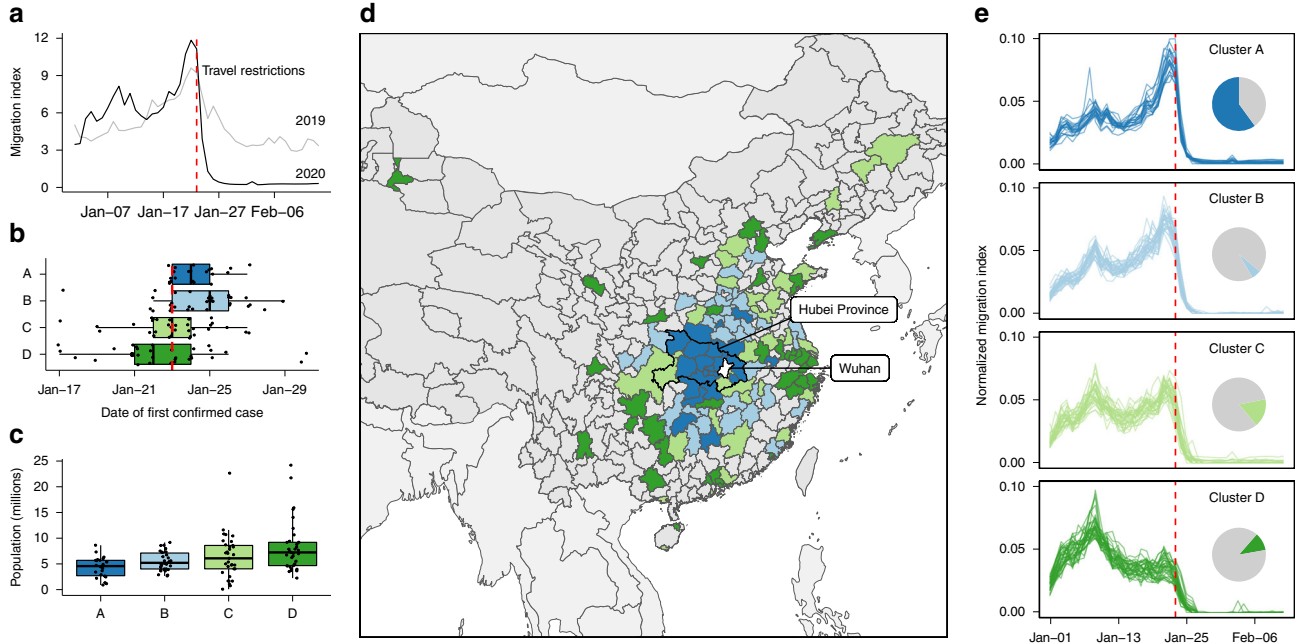

**Fig. 1 Travel patterns between Wuhan and other connected prefectures.** The identified patterns of outbound travel from Wuhan: (**a**) the daily total outbound travel from Wuhan in 2019 and 2020; (**b**) timing of first case detection stratified by clusters of similar time series; (**c**) distribution of resident population sizes of individual prefectures (points); (**d**) map of prefectures and province-level cities showing the spatial distribution of timeseries clusters; (**e**) outbound travel trends from Wuhan to the most connected prefectures in China, stratified by clusters with similar time series. The clusters are defined by k-means clustering of the timeseries of outbound travel volume (see "Methods" section). For clusters in panels **b** and **c**, n = 22 (Cluster A), 34 (Cluster B) 33 (Cluster C), 36 (Cluster D). Boxplots in panels **b** and **c** display Median, IQR, and whiskers +/− 1.5 times IQR. The timeseries have been normalised by the total flow of each, to allow comparison of the profile. Inset pie charts show the total travel flux out of Wuhan prefecture by destinations in each cluster. The red dashed lines in panels **a**, **b** and **e** mark the beginning of LNY holidays. The colours in panels **b**, **c**, **d** and **e** indicate cluster membership (Cluster A, B, C or D).

between-year difference; and (2) an anomaly-based metric Eq. (4) that captures the deviation observed in 2020 compared to 2019. We found that although there is evidence of an increase in outbound travel from Wuhan during this period, a similar increase was also observed in many other prefectures. Wuhan was ranked 46 (top 13%) and 88 (top 24%) of 305, by the two metrics for the change in flow (Supplementary Fig. 12).

**Movement patterns across China.** We explored the existence of hierarchical patterns of movement between differently-sized prefectures in mainland China, in an effort to understand differences in the connectivity between more and less populated prefectures during heightened travel during the LNY holidays. We divided prefectures and province-level cities into four population quartiles (i.e., Low (2000 to 1.44 million residents), Medium-low (1.45 to 2.96 million residents), Medium-high (2.98 to 4.90 million residents) and High (4.92 to 24.20 million residents). We found that the trends of inbound and outbound travel volume over time were relatively consistent across population quartiles (Supplementary Fig. 13). The flow between all pairs of quartiles, measured in Baidu's migration index, increased prior to LNY and dropped sharply after Wuhan's *cordon sanitaire*, with an increase in within-quartile flow following 23rd January for all quartiles. However, the underlying composition of these in- and outbound travel flows differed substantially by population quartile (Fig. 2).

Before LNY, all regions saw increased inbound travel from highly populated prefectures (Fig. 2a–d). These changes were more marked in prefectures of lower population sizes. After LNY, the contribution to inbound travel by prefectures in the middle

quartiles stabilised at higher levels compared to pre-LNY. As the volume of inbound travel recovered through February (Supplementary Fig. 15), the relative proportion of travellers from the most populated quartiles remained low. For outbound travel, a higher proportion of travellers from the most populated prefectures travelled to the middle quartiles before LNY, and a higher proportion from medium-sized prefectures travelled to low-population prefectures (Fig. 2e–h). Travel volumes and distance patterns in Beijing, Shanghai, and Guangzhou began to return to normal more quickly than in Wuhan, and outbound travel generally recovered more after LNY (Supplementary Fig. 14).

This analysis of origin or destination locations revealed diverging hierarchical effects, rather than a simple cascading flow of travellers from larger to smaller population prefectures. Travellers from large prefectures more often travelled to other large or medium size prefectures; travellers from medium and small prefectures more often travelled between medium and small prefectures. Holiday travel immediately preceding LNY can be considered an indicator of long-term migration in China, as people travel back along their long-term migration route temporarily to visit family. The patterns we observed are consistent with the migration step effect along the urban hierarchy, in which geographic regions of similar population size exchange members more often[15,16]. The divergence in hierarchical flow between high and low population prefectures means that middle population prefectures could play a key role in limiting the spread of COVID-19 to prefectures with fewer residents. Non-pharmaceutical interventions could target these medium-sized prefectures to prevent epidemics from reaching the relatively rural parts of China.

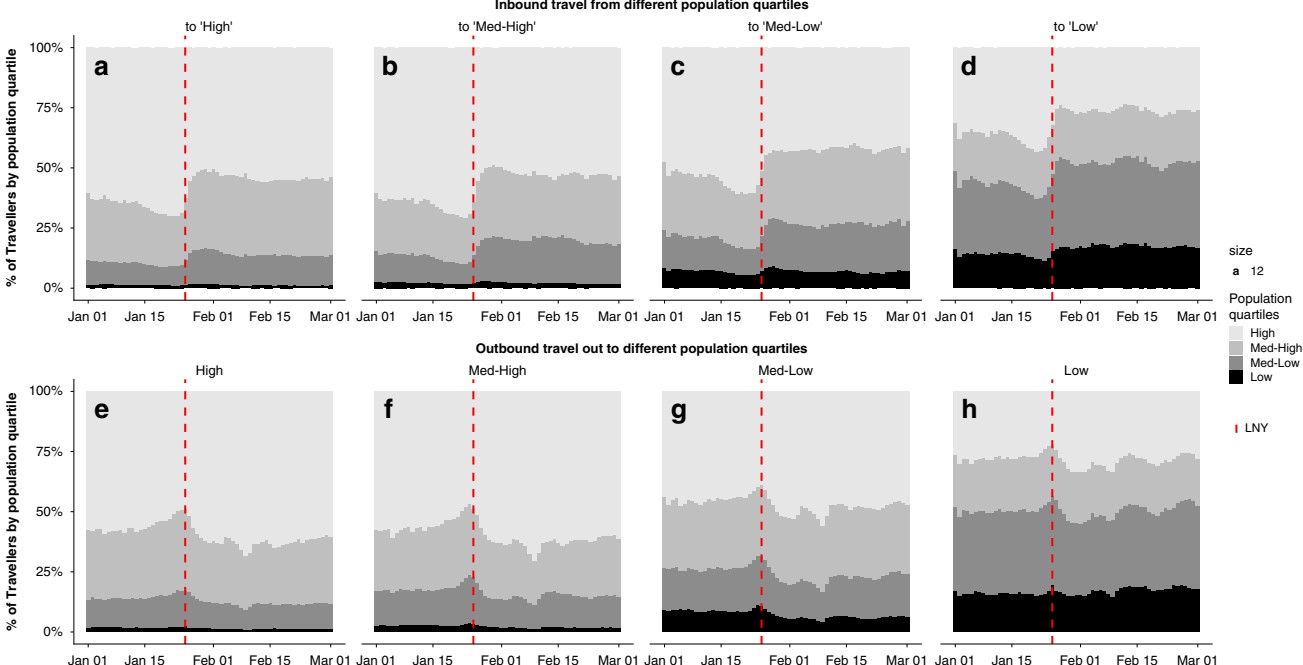

**Fig. 2 Contribution of each population quartile to inbound and outbound travel.** Shading marks the population quartile with highest population quartiles in the lightest shade. The red dashed line shows the first day of LNY (25th January 2020). The figure displays the proportion of travel from all quartiles to High (**a**), Medium-high (**b**), Medium-low (**c**) and Low (**d**) population quartiles, as well as the proportion of travel to all quartiles originating from High (**e**), Medium-high (**f**), Medium-low (**g**) and Low (**h**) population quartiles.

**Healthcare capacity and COVID-19-related healthcare pressure**. Before the LNY, the move away from larger population centres was also a move away from high healthcare capacity, measured by the number of Grade II and III hospitals per 100,000 residents (Fig. 3). Prefectures with higher healthcare capacity had more outgoing than incoming travellers, and after LNY, travellers gradually returned to high healthcare capacity settings, but the overall geographic distribution of residents had not recovered to its pre-LNY conditions by 1st March 2020 (Fig. 3a). This pattern persisted when we used an alternative healthcare capacity measure of the number of Grade II and III hospitals without adjusting for background population size (Supplementary Fig. 16).

The movement observed was associated with COVID-19-related healthcare pressure (see "Methods" section), a measure of confirmed cases compared with healthcare capacity (Fig. 3b). From the week before LNY to two weeks after, locations with low healthcare capacity experienced significantly higher pressure compared to locations with high healthcare capacity. Therefore *Chunyun* not only increased the chance of infection along mobility networks, but also shifted healthcare pressure caused by COVID-19 to regions with low healthcare capacity, an effect seen in other countries and during natural disasters[17–19]. Using the alternative healthcare capacity measure that considers number of hospitals only, we found similar relative associations (Supplementary Fig. 16).

**Changes in overall travel network structure**. In order to understand broad changes in the Chinese transportation network, we identified communities of highly connected prefectures and assessed the change in these communities during LNY and the introduction of local interventions in Chinese prefectures. We determined the community structure of the local travel network by calculating the daily modularity, Q, of the directed network[20] from 1st January to 1st March 2020. Each community (or module) has more connections within vs. between communities, and modularity is one method for measuring community structure in

networks. The changing modularity provides a holistic view of transport throughout the country, highlighting macroscopic changes in the network, e.g., rerouting behaviour or increased linkages between new prefectures, as the movement network adjusted to travel restrictions in Wuhan.

Preceding the implementation of travel restrictions, there was a stable pattern of communities connected to large cities, with significant flows between communities (Fig. 4). A lower Modularity value (Q) indicates weaker connections between prefectures within a community, or higher volumes of travel between different communities, rather than within the same communities. The low values of Q preceding LNY indicate a high volume of travel between communities, with increased interconnection of the movement network. Early January before LNY represents typical travel in China with flow between major population centres. During this period, travel within China was generally structured into well-defined communities, with high modularity, Q (Fig. 4, time point 1). Major cities had consistent, distinct communities which remained fairly steady even as outflows began to increase from major cities for LNY (Fig. 4, time point 2; see Supplement 6 for full time series).

Immediately following the implementation of travel restrictions, we identified a marked peak in modularity where the Q-value for Wuhan City increased, indicating that it temporarily became more integrated into the travel network (Fig. 4a, time point 3). This increase in modularity indicated relatively more connectivity between Wuhan and other communities, although there was decreased flow, so the actual number of travellers was much lower. This could also reflect the large movement of medical and other resources to Wuhan following the implementation of restrictions[21].

Overall connectivity decreased across China after the cordon sanitaire in Wuhan (Fig. 4, time point 4). This coincided with the implementation of disease control interventions in other prefectures, and a decrease in travel following LNY. Consistent with a country-wide policy of restricted movement, we did not

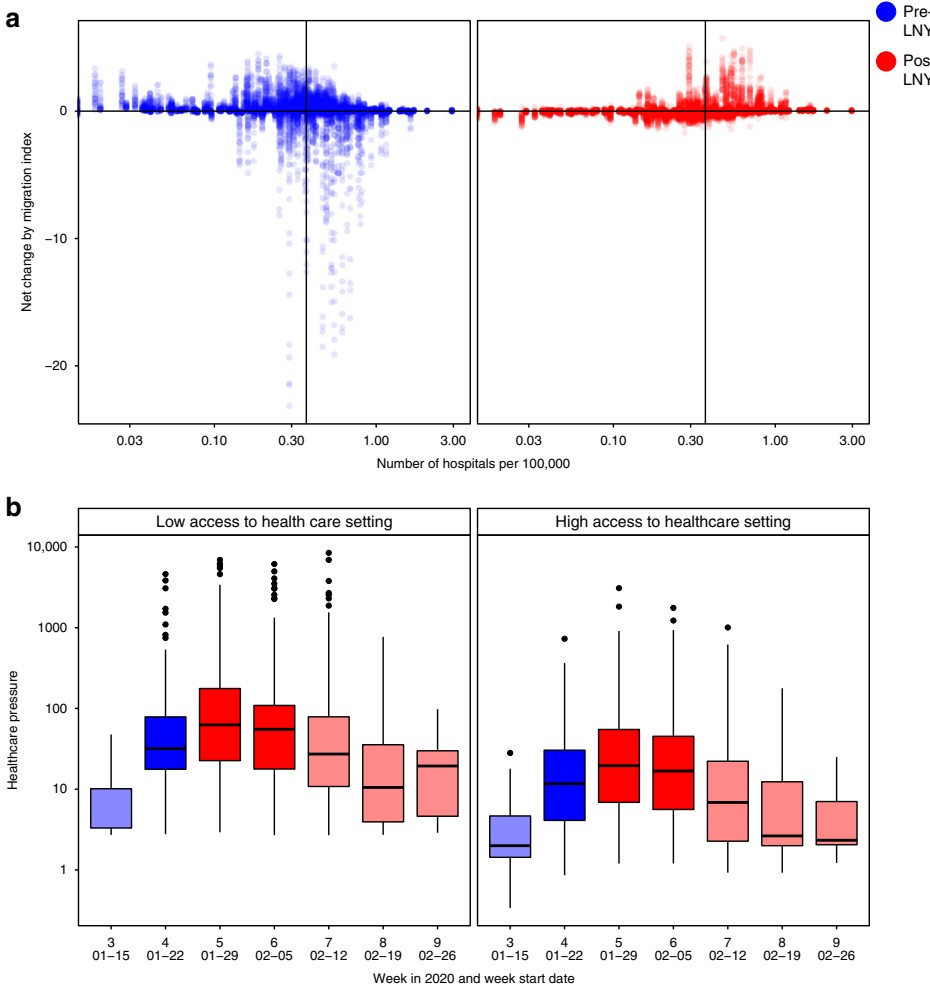

**Fig. 3 Healthcare service availability and COVID-19-related healthcare pressure. a** The changes in traveller volume before (blue) and after (red) LNY. Net change is defined as inbound migration index minus outbound migration index. Thus, a negative change indicates more travellers leave than arrive while a positive value indicates more travellers arrive than leave. A solid line indicates the median level of healthcare capacity. **b** The changes in the healthcare pressure ($\log_{10}$ scale) related to COVID-19 each week in low and high healthcare capacity prefectures. Healthcare capacity is measured by the number of hospitals per 100,000 residents ($n_{low} = 157$, $n_{high} = 153$). Healthcare pressure is measured by confirmed COVID-19 cases divided by healthcare capacity. Darker shade represents weeks when low healthcare capacity settings experienced significantly higher pressure than high healthcare capacity settings; lighter shade represents when differences are not statistically significant based on Mann–Whitney $U$ test (5% type I error rate). The comparison for week 7 has $p$-value = 0.06. The boxplots in panel **b** display Median, IQR and whiskers $+/-$ 1.5 times IQR.

find large rerouting or the increasing importance of other transport connections after the restrictions in Wuhan. This is critical as countries attempt to determine the efficacy of large-scale movement restrictions.

## Discussion

The cordon sanitaire in Wuhan was an intensive travel restriction that completely stopped all non-essential incoming and outgoing traffic. Previous studies have demonstrated that it may have had low effectiveness in preventing or delaying transmission to other regions of mainland China during the early phase of the COVID-19 pandemic[7,22]. There is however potential for infectious disease control and prevention, especially when timeliness and the necessary scope of restrictions can be achieved[23]. Travel restrictions will likely continue to be considered an important infectious disease intervention option against COVID-19 during the pandemic, and better understanding the mechanisms in play at different stages of travel restrictions is crucial to effective implementation.

We found a limited relationship between spatial proximity and epidemic spread where larger, distant populations detected their first COVID-19 cases earlier than smaller locations that are closer to Wuhan. We also observed a hierarchical divergence of movement between prefectures of different populations sizes, with larger prefectures more connected to other large prefectures, and smaller prefectures more connected to other small prefectures. Due to the highly connected modern mobility network, spatial proximity is not the only measure for closeness between two cities[24] We found a limited relationship between spatial proximity and epidemic spread where larger, distant populations detected their first COVID-19 cases earlier than smaller locations that are closer to Wuhan. Due to the highly connected modern mobility network, spatial proximity is not the only measure for closeness between two cities[24]. While planning for travel restrictions, either domestic or international, it may be worthwhile to consider other functional connectivity measures, such as human mobility studied here. Although outbreaks may appear to have single source location in the beginning, such as the case in Europe[25], focussed travel restrictions around epicentres and their

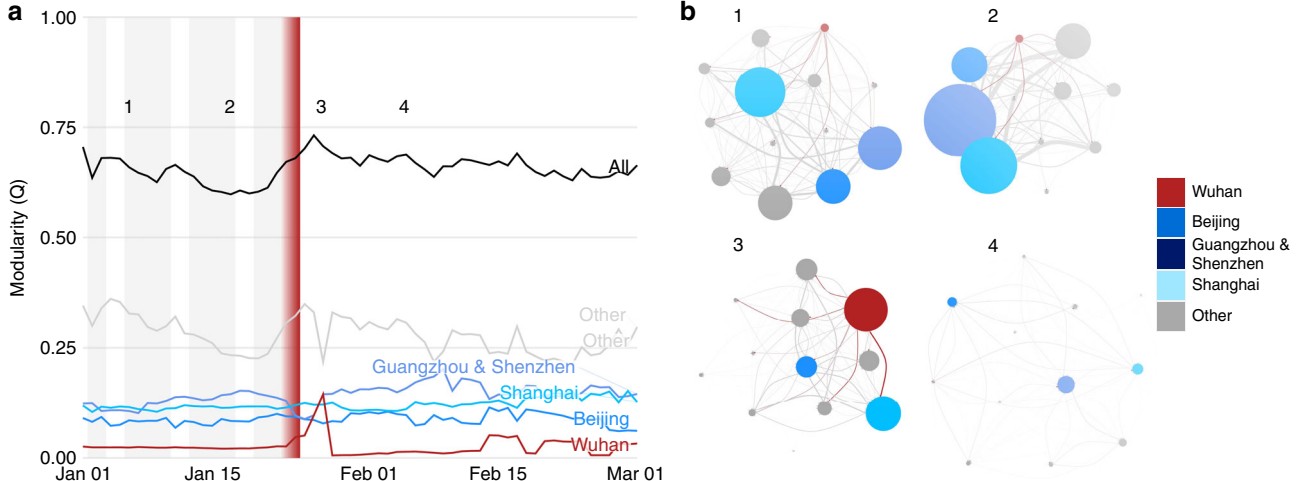

**Fig. 4 Community structure of the movement network.** Community structure is measured through modularity, a metric defining the strength of connections within vs. between communities. The members of communities are determined by the Leiden algorithm[20]. **a** The time series of total and sub-community modularity, and (**b**) snapshots of the community networks on days before and after the *cordon sanitaire*. The communities for Wuhan (red) and several major cities (blues) are highlighted in **a** and **b** in both the community and edges between communities. The time series shows working week (grey bars; missing after nation-wide social distancing measures), as well as the initiation and enforcement of restrictions in Wuhan (red gradient) over 23rd to 24th January. The communities (circles) are sized according to within-community migration index, while their connections are sized according to their between-community migration index.

immediate geographic surroundings may lead to missed opportunities for epidemic control.

The timing of LNY and the initial stage of the COVID-19 epidemic makes it difficult to untangle regular holiday travel from travel in response to the outbreak or to impending travel restrictions. The increased outflow from Wuhan that we observed was not unique to the city, as similar patterns of outflow were observed in a large number of other prefectures, and so likely represents increased holiday travel. We therefore did not find evidence of an association between the announcement of the cordon sanitaire and the number of outbound travellers leaving Wuhan. Data from other countries not confounded by holiday travel (e.g., France[26]) may yield insights on public responses to travel restrictions. In addition, although the overall number of travellers leaving Wuhan was not exceptionally high before LNY, the composition of travellers may have changed, such as a shift from business to family travel, which could contribute to the spread of COVID-19 and could have implications for healthcare demand in destination locations[27]. Finer resolution mobility data, including traveller characteristics such as age and occupation, could improve our understanding of the potential outbreak risk and the likely impacts of different interventions in the future.

Human mobility during *Chunyun* was marked by the general trend of people leaving large population centres for less populated locations. This is a move by the population away from locations with high healthcare capacity. During the peak of the epidemics in mainland China, areas with low healthcare capacity experienced significantly higher healthcare pressure related to COVID-19 compared to elsewhere. Temporarily mobilising resources such as medical personnel and equipment could aid epidemic control in places receiving a higher-than-normal number of travellers from places with potentially high COVID-19 prevalence, and thus could be evaluated as a potential public health intervention under similar circumstances[28].

The structure of the overall transportation network in China did not demonstrate compensatory responses to the *cordon sanitaire*. There was a brief alteration of the network structure immediately following the restrictions, before the network settled quickly back into the same relatively stable communities that

existed before the restrictions, albeit at markedly lower flow. This implies that the overall transportation network did not undergo structural reorganisation as a result of Wuhan's cordon sanitaire and other regional travel restrictions. Short-term travel restrictions may therefore not incur lasting impacts on the mobility network, but assessing long-term impacts will require longer time-series analyses.

Mobility data from Baidu Huiyan has some limitations. For example, travel volumes were collected on an eight-hourly basis between each pair of prefectures and then aggregated to day-level and prefecture-level, which does not allow analysis of trips longer than a day. In a country the size of China, such trips may be relatively frequent. Pairwise travel patterns before 1 January 2020 are not available, which makes it challenging to determine baseline travel patterns. In addition, movement patterns from Baidu Huiyan reflect the movement of Baidu users, which may be a non-random subset of the general population in mainland China[29].

This study analysed the human mobility patterns around China during different stages of the local COVID-19 epidemics, from early *Chunyun* to Wuhan's cordon sanitaire and other travel restrictions. Using a range of techniques, we assessed the patterns of movement specific to Wuhan and the characteristics of the travel network throughout China considering the implications of changing travel patterns on the spread of COVID-19. We also explored the impact of travel patterns on Chinese prefectures, assessing the changes in healthcare pressure due to varying patterns of human mobility typically associated with LNY, which coincided with the early stages of the COVID-19 pandemic. Many countries have now implemented similar travel restrictions to reduce disease transmission. Understanding the implications of travel patterns before, during, and following travel restrictions is valuable for informing public health interventions, surveillance, and healthcare demand planning globally.

## Methods

**Geographic information.** The geographic unit of analysis in this study is prefecture, which is administrative level two in mainland China, just below the province (level one). There are currently more than 360 prefecture-level units in

China. However, the four provincial level cities (Beijing, Tianjin, Shanghai and Chongqing) are exceptions. They do not have a level two unit - level one directly manages level three administrative units (i.e., counties) in these locations[30]. In this study, we analysed these province-level cities with prefectures for spatial completeness.

**Mobility data.** The mobility data is publicly available through Baidu Huiyan[10], a web service that supports government agencies and businesses with big-data spatio-temporal analytics. Estimates are based on over 120 billion location-based service (LBS) enquiries each day from over 1.1 billion mobile devices, while taking into consideration more than 1.5 billion points of interests (POI). We obtained two variables directly from Baidu Huiyan: overall migration index (specific to each prefecture) and percentage of travellers arriving in or leaving specific locations (specific to each pair of prefectures). Note that migration index is a relative measure of the magnitude of human mobility, scaled relative to the total volume of movement across the network. Baidu movement flow index is collected in 8-h windows and is provided as origin-destination flows between pairs of prefectures. We further processed these data to produce symmetrical matrices of daily travel between all Chinese prefectures.

We calculate the volume of human mobility between each pair of prefectures on each day between 1st January 2020 and 1st March 2020 using the following equation:

$$T_{ij,t} = F_{i,\text{outbound},t} \times p_{ij,\text{outbound},t} \tag{1}$$

where, for a given day, $T_{ij,t}$ is the volume of mobility from location $i$ to location $j$ on day $t$, $F$ is the overall Baidu migration index with direction (inbound or outbound) at location $i$, and $p_{ij,\text{outbound}}$ is the proportion of all outbound travel that originated in $i$ and ended in $j$. References to inbound and outbound travel are made in regards to a specific origin or destination location. We further validated this measure by assuming that inbound and outbound were equal, as:

$$T_{ij,t} = F_{i,\text{outbound},t} \times p_{ij,\text{outbound},t} = F_{j,\text{inbound},t} \times p_{ij,\text{inbound},t} \tag{2}$$

where $p_{ij,\text{inbound},t}$ is the proportion of all inbound travel that end in $j$ and originate in $i$. Note that $p_{ij,\text{outbound},t}$ and $p_{ij,\text{inbound},t}$ are only available for the top 100 connected prefectures. In other words, $p_{ij,\text{outbound},t}$ is only available for the top 100 destinations originating in $i$; $p_{ij,\text{inbound},t}$ is only available for the 100 origins with the most travellers to $j$. We were not able to validate for $T_{ij,t}$ in the cases where $p_{ij,\text{outbound},t}$ and $p_{ij,\text{inbound},t}$ are not simultaneously available. Using data from Baidu Huiyan, we created a symmetric, $366 \times 366$ connectivity matrix for each day between 1 January 2020 and 1 Mar 2020 (61 days).

**Demographic and healthcare system data.** The 2018 population sizes were retrieved from the China Statistics Yearbook[31]. This metric accounts for migrant population, and thus is expected to may fluctuate during the holiday seasons. The geographic boundaries of prefectures and province-level cities were obtained from the Institute of Geographic Sciences and Natural Resources Research (Chinese Academy of Sciences)[32]. The original source of daily confirmed incidence is the COVID-19 dashboard published by DXY.cn, which updates in near-real time based on government press releases[33]. In addition, the package 'nCoV2019'[34] and 'DXY-COVID-19-Crawler'[35] have reduced the time required for data gathering and data cleaning. Records of first case arrivals were cross-checked with news articles also found throughon DXY.cn[33]. Information on the Grade II and III hospitals in China was retrieved from the National Health Commission[36] and was then geo-referenced using the non-commercial Amap API[37].

**Time series analysis and surge evaluation.** The patterns of movement out of Wuhan between 1st and 23rd January were analysed using cluster analysis of the magnitude-normalised timeseries of outflow over time. Outflow timeseries were selected using a threshold of journeys with an average flow index greater than 0.005 for the entire period. This threshold removed prefectures with negligible connectivity with the origin.

In order to characterise the shape of the outflow from Wuhan, rather than the magnitude of certain outflows, we calculated the normalised flow, $N$, between origin ($i$) and destination ($j$) prefectures on each day ($t$), by dividing the outflow measured by the travel index $T$, by the total movement between the 1st and 23rd January 2020, as:

$$N_{ij,t} = \frac{T_{ij,t}}{\sum_{t=1}^{t=23} T_{ij,t}} \tag{3}$$

We classified the time series using k-means clustering with four clusters[38]. The number of clusters was chosen using a plot of average silhouette width against number of clusters, for between 4 and 12 clusters. The silhouette width decreased significantly at four clusters, and a similar number of time series were allocated to each cluster (Supplementary Figs. 1–5). Furthermore, when using a greater number of clusters, we observed the same four overall temporal patterns with smaller differences between time series defining each cluster. We also observed an increasingly large number of clusters containing a small number of time series. Plots of the time series clustered using 2, 3, 4, 5 and 6 clusters are included in

Supplementary Figs. 1–5. K-medioids, and Agglomerative Clustering were also explored as alternatives to K-means clustering. The different clustering methods did not result in substantial differences and identified similar patterns among outflow time series.

We quantified the peak outflow from each prefecture in the five-day window before LNY (i.e., two to seven days before LNY, zero to five days before the *cordon sanitaire*). We used two parameters to characterise the magnitude of the change in outflow in 2020 compared to 2019 in each prefecture $i$:

$$V_{1,i} = \frac{\text{mean}\left(F_{i,\tau_{2020}}\right)}{\text{mean}\left(F_{i,\tau_{2019}}\right)} - 1 \tag{4}$$

$$V_{2,i} = \frac{\text{mean}\left(F_{i,\tau_{2020}}\right) - \text{mean}\left(F_{i,\tau_{2019}}\right)}{\text{std}\left(F_{i,\tau_{2019}}\right)} \tag{5}$$

where $F_{i,\tau}$ is the total Baidu outflow from prefecture $i$ in the time period $\tau$. $\tau$ in 2019 corresponds to 29 January–3 February 2019, and in 2020 corresponds to 18 January–23 January. The dates are different each year because they are aligned to the date of LNY in 2019 and 2020.

*The Relationship between First Case Detection and Cluster Membership*: We explored the association between average population size and first case detection for prefectures in each cluster. There is potential confounding due to surveillance bias such that larger prefectures may detect COVID-19 cases earlier due to better public health infrastructure resulting in earlier and greater use of diagnostic tests. However, there is no intuitive indicator that can capture surveillance efforts, and therefore we used population size as a proxy for surveillance effort. The implicit assumption is that places with larger populations are more equipped for detecting COVID-19, which is supported because early testing capacity relied on biosafety level 2+ laboratories, which are only found in large hospitals and universities[39].

We adjust for this potential confounding effect using a linear regression model:

$$\text{Detection date} \sim \beta_o + \beta_1 \times \text{pop} + \beta_2 \times (\text{cluster membership}) \tag{6}$$

where pop represents the prefecture level population size as of 2018 and (Cluster membership) is a nominal unordered categorical variable with levels A through D.

**Assessing the healthcare capacity and COVID-19-related healthcare pressure.** In this study, prefecture-level healthcare capacity was measured by the number of Grade II and III hospitals per 100,000 residents. In mainland China, Grade II and III hospitals have 100–499 or 500+ hospital beds, respectively, and are equipped with ventilators[40]. Thus, they are more important compared to community hospitals and clinics for COVID-19 management. Healthcare capacity in prefecture $i$ ($\text{HC}_i$), therefore, can be expressed as:

$$\text{HC}_i = \frac{n_{\text{hospital},i}}{\text{Pop}_{\text{residential},i}} \tag{7}$$

where $n_{\text{hospital},i}$ is the number of Grade II and III hospitals in prefecture $i$, and $Pop_{\text{residential},i}$ is the residential population of prefecture $i$. We use the size of the residential population in 2018 from the China Statistics Yearbook[29]. $\text{HC}_i$ is stratified into high and low by taking the upper and lower 50% of prefectures with available data. Note that this metric cannot accurately reflect the prefecture level population sizes during LNY due to population movement. For example, the residential population size of Beijing is approximately 22 million, and over 10 million left the city for LNY[41].

Healthcare pressure in prefecture $i$ during week $w$ ($\text{Hp}_{i,w}$) was calculated by dividing weekly confirmed COVID-19 cases[31] by the healthcare capacity:

$$HP_{i,w} = \frac{n_{\text{confirmed},w}}{HC_i} \tag{8}$$

where $n_{\text{confirmed},w}$ is the number of confirmed COVID-19 cases during week $w$.

We also performed a sensitivity analysis on the metric and considered an alternative measure of healthcare capacity that did not adjust for the background residential population sizes:

$$HC_i = n_{\text{hospital},i} \tag{9}$$

The distributions of healthcare pressure were non-Gaussian. We therefore used non-parametric one-tailed Mann–Whitney $U$ tests to compare the differences of healthcare pressure between low and high healthcare capacity settings. The null hypothesis was that the healthcare pressures in low healthcare capacity settings are comparable to that in high healthcare capacity settings; the alternative hypothesis was that healthcare pressures in low healthcare capacity settings are higher than those in high healthcare capacity settings (n1 = 157, n2 = 153). This test was repeated for each week from week three to nine (i.e., starting on 15 Jan 2020). Results were verified using two-tailed Mann–Whitney $U$ tests, as well as one-tailed Mann–Whitney $U$ tests with the opposite null hypotheses.

**Network analysis.** Using the weighted movement flows between locations, we calculated community structure in the network using the Leiden algorithm[20]. The Leiden algorithm maximises the modularity, $Q$, on directed, weighted, time sliced

networks with an inter-slice weighting of $10^{-5}$, which is the order of magnitude of minimum intra-slice weight across all times[42]. Modularity is a metric of within-community vs. between-community connectivity, and the algorithm detects communities by optimising the within vs. between, thereby assigning nodes to communities. Using the community structure from this algorithm, we identified the relative contributions to modularity, $Q$, of 4 key communities: the community containing Wuhan prefecture, and then the communities of four other major cities in China: Beijing, Shanghai, Guangzhou, and Shenzhen. The latter two were always assigned to the same community and are marked together in Fig. 4. We presented 4 snapshots of communities in the travel network, but all are shown in Supplementary Fig. 17, and the spatial locations of those networks in Supplementary Fig. 18.

**Distance Kernels**. To determine how the relationship between distance and travel flow changed over *Chunyu*n and in response to the *cordon sanitaire*, we calculated the frequency of journeys of at least distance $n$ kilometres, for $n$ up to the maximum distance 4185 km, on each day of the study period. These plots are shown for Beijing, Guangzhou, Shanghai and Wuhan, for both inflow and outflow in Supplementary Fig. 14.

**Sensitivity analyses**. We repeated clustering of temporal traveller flow time series to validate the method for assessing travel flux out of Wuhan between January 1st and January 23rd. Employing the same method of thresholding prefectures with little connectivity, the volume of travel to individual destination locations over time were normalised by dividing by the total flow along each route in the period. These normalised time series were then clustered using the same k-means clustering procedure discussed above. The number of clusters was determined using a silhouette plot in order to isolate the dominant temporal patterns of traveller movement to individual destinations.

**Reporting summary**. Further information on research design is available in the Nature Research Reporting Summary linked to this article.

## Data availability
All data used in this study are publicly available. Analyses were conducted using a symmetrical matrix of movement flux data collected from the Qianxi Baidu 2020 movement map: https://qianxi.baidu.com/. The processed movement dataset is included in a publicly accessible repository: https://github.com/yangclaraliu/pandemic_travel_china. This research also relies on Covid-19 case count data and prefecture level hospital resource data from China, C. D. C. Public Health Science Data Centre, as well as prefecture level population data from China Statistical Yearbook 2018. Covid-19 case count data, prefecture level hospital resource data, and prefecture level population data require applications for use.

## Code availability
Code is publicly available in a Github repository, https://github.com/yangclaraliu/pandemic_travel_china.

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

## Acknowledgements
The following funding sources are acknowledged as providing funding for the named authors. This research was partly funded by the Bill and Melinda Gates Foundation (INV-003174: Y.L.; NTD Modelling Consortium OPP1184344: C.A.B.P.). DFID/Wellcome Trust (Epidemic Preparedness Coronavirus research programme 221303/Z/20/Z: C.A.B.P.). This project has received funding from the European Union's Horizon 2020 research and innovation programme-project EpiPose (101003688: Y.L.). This research was partly funded by the Global Challenges Research Fund (GCRF) project 'RECAP'

managed through RCUK and ESRC (ES/P010873/1: C.I.J.). HDR UK (MR/S003975/1: R.M.E.). This research was partly funded by the National Institute for Health Research (NIHR) using UK aid from the UK Government to support global health research. The views expressed in this publication are those of the author(s) and not necessarily those of the NIHR or the UK Department of Health and Social Care (16/137/109: B.J.Q., C.D., Y.L.). UK DHSC/UK Aid/NIHR (ITCRZ 03010: H.P.G., C.G.). UK MRC (MC_PC 19065: R.M.E.). The following authors were part of the Centre for Mathematical Modelling of Infectious Disease 2019-nCoV working group. Each contributed in processing, cleaning and interpretation of data, interpreted findings, contributed to the manuscript, and approved the work for publication: Sophie R Meakin, Petra Klepac, Samuel Clifford, Rachel Lowe, Sam Abbott, Joel Hellewell, Jon C Emery, Akira Endo, Georgia R Gore-Langton, Adam J Kucharski, Simon R Procter, Amy Gimma, Sebastian Funk, Kathleen O'Reilly, Oliver Brady, Thibaut Jombart, Mark Jit, Kiesha Prem, W John Edmunds, Nikos I Bosse, Fiona Yueqian Sun, C Julian Villabona-Arenas, Gwenan M Knight, Nicholas G. Davies, Matthew Quaife, James W Rudge, Damien C Tully, David Simons, Alicia Rosello, Quentin J Leclerc, Emily S Nightingale, Eleanor M Rees, Anna M Foss, Katherine E. Atkins, Kevin van Zandvoort, James D Munday, Arminder K Deol, Timothy W Russell, Megan Auzenbergs, Stefan Flasche, Rein M G J Houben, Graham Medley, Stéphane Hué. The following funding sources are acknowledged as providing funding for the working group authors. Alan Turing Institute (A.E.). BBSRC LIDP (BB/M009513/1: D.S.). This research was partly funded by the Bill and Melinda Gates Foundation (INV-001754: M.Q.; INV-003174: KP; INV-003174; This research was partly funded by the National Institute for Health Research (NIHR) using UK aid from the UK Government to support global health research. The views expressed in this publication are those of the author(s) and not necessarily those of the NIHR or the UK Department of Health and Social Care: M.J.; NTD Modelling Consortium OPP1184344: G.M.; OPP1180644: SRP; OPP1183986: ESN; OPP1191821: KO'R, MA). DFID/Wellcome Trust (Epidemic Preparedness Coronavirus research programme 221303/Z/20/Z: K.v.Z.). DTRA (HDTRA1-18-1-0051: J.W.R.). Elrha R2HC/UK DFID/Wellcome Trust/NIHR (K.v.Z.). ERC Starting Grant (#757688: C.J.V.A., KEA; #757699: J.C.E., R.M.G.J.H.; 757699: M.Q.). This project has received funding from the European Union's Horizon 2020 research and innovation programme-project EpiPose (101003688: K.P., P.K., W.J.E.). This research was partly funded by the Global Challenges Research Fund (GCRF) project 'RECAP' managed through RCUK and ESRC (ES/P010873/1: A.G., T.J.). Nakajima Foundation (A.E.). NIHR (16/137/109: F.Y.S.; Health Protection Research Unit for Immunisation NIHR200929: N.G.D.; Health Protection Research Unit for Modelling Methodology HPRU-2012-10096: T.J.; PR-OD-1017-20002: A.R.). NIHR200929; European Commission (101003688: M.J.). Royal Society (Dorothy Hodgkin Fellowship: R.L.; RP\EA \180004: P.K.). UK MRC (LID DTP MR/N013638/1: E.M.R., G.R.G.L., Q.J.L.; MR/P014658/1: G.M.K.). Authors of this research receive funding from UK Public Health Rapid Support Team funded by the United Kingdom Department of Health and Social Care (T.J.). Wellcome Trust (206250/Z/17/Z: A.J.K., T.W.R.; 206471/Z/17/Z: O.J.B.; 208812/Z/17/Z: S.C., SFlasche; 210758/Z/18/Z: J.D.M., J.H., N.I.B., S.A., S.Funk, S.R.M.). No funding (A.K.D., A.M.F., D.C.T., S.H.).

## Author contributions
H.G., Y.L., R.M.E., conceived the methods in the study. H.G., Y.L., C.I.J. and C.A.B.P. implemented the analysis and generated the figures with input from B.J.Q., C.D., R.M.E. and C.G. All authors interpreted the findings and prepared the manuscript. All authors reviewed the manuscript and approved the final version for submission.

## Competing interests
The authors declare no competing interests.

## Additional information

## LSHTM CMMID COVID-19 working group

David Simons[2], Amy Gimma[2], Quentin J. Leclerc[2], Megan Auzenbergs[2], Rachel Lowe[2], Kathleen O'Reilly[2], Matthew Quaife[2], Joel Hellewell[2], Gwenan M. Knight[2], Thibaut Jombart[2], Petra Klepac[2], Simon R. Procter[2], Arminder K. Deol[2], Eleanor M. Rees[2], Stefan Flasche[2], Adam J. Kucharski[2], Sam Abbott[2], Fiona Yueqian Sun[2], Akira Endo[2], Graham Medley[2], James D. Munday[2], Sophie R. Meakin[2], Nikos I. Bosse[2], W. John Edmunds[2], Nicholas G. Davies[2], Kiesha Prem[2], Stéphane Hué[2], C. Julian Villabona-Arenas[2], Emily S. Nightingale[2], Rein M. G. J. Houben[2], Anna M. Foss[2], Damien C. Tully[2], Jon C. Emery[2], Kevin van Zandvoort[2], Katherine E. Atkins[2], Alicia Rosello[2], Sebastian Funk[2], Mark Jit[2], Samuel Clifford[2] & Timothy W. Russell[2]

