## [Peer Review File · Nature Communications]

REVIEWER COMMENTS

Reviewer #1 (Remarks to the Author):

Recommendation: Minor revision

Key Results:

In this manuscript, Gibbs et al analyze travel data from Baidu Huiyan to describe travel patterns in China during the emergence of the COVID-19 epidemic in January and February 2020. Changes in travel flows and travel network structure are analyzed in the context of key events over the relative time period, such as the Lunar New Year and the implementation of a cordon sanitaire.

The key analyses and findings of this paper include:

- An analysis of temporal patterns of travel between Wuhan and other prefectures in China. In comparison to 2019, the authors identify a 2020 peak in outbound travel in early January that is difficult to definitively explain, as well as a higher peak just preceding the start of the lunar new year. The authors do not find evidence for an anecdotally-reported, anomalous, Wuhan-specific increase in outbound travel just prior to or after the announcement of the cordon sanitaire.
- A cluster analysis of patterns of outbound travel from Wuhan. The authors use k-means clustering to create four groups of destination prefectures according to temporal trajectories of outbound travel from Wuhan. Combining these groupings with an analysis of timing of first case detection, this analysis suggests that travel to more spatially distant, more populous prefectures was relatively higher in early January compared to just before LNY, and the timing of first case detection was slightly earlier (~1 day) in these destination prefectures. Travel to more spatially proximate prefectures was most common overall, peaked just prior to the LNY, and was associated with slightly later detection of first cases.
- An analysis of movement patterns between prefectures grouped by population quartile. Pre-LNY movement indicated increased movement out of highly populated prefectures to other locations, particularly those with lower population sizes. After the cordon sanitaire, overall population flows decreased markedly, but relatively more travel originated from less-populated prefectures.
- An analysis of healthcare capacity and population size. During the pre-LNY period, travel from areas with higher to areas with lower healthcare capacity was common. The authors generate a measure of healthcare pressure using confirmed cases and capacity and suggest that pre-LNY travel shifted healthcare pressure to places that had low baseline capacity.
- An analysis of the prefecture-level travel network structure and the evolution of modularity over time. This analysis suggests that travel flow and connectivity between prefectures decreased broadly throughout China following the cordon sanitaire. There was a brief restructuring immediately following the cordon sanitaire, with Wuhan assuming increased importance in the network – perhaps due to outbreak response activities, the authors suggest – but a rapid reversion to the pre-LNY state and no persistent substantial restructuring of the travel network.

Novelty and interest

As the authors appropriately note in the introduction, Baidu mobility data has been used from in several other analyses of COVID in China. For example, Lai et al. (Nature 2020) use Baidu mobility data as an input into a travel network-based SEIR model simulating COVID spread; they briefly describe the daily outflow from 340 cities within mainland China but their main focus is in modeling of the epidemic. They do not assess travel patterns in early January 2020. Chinazzi et al (Science 2020) similarly use Baidu mobility data as an input into an individual-based stochastic spatial epidemic model. This analysis uses Baidu mobility indices at the province level to parameterize the metapopulation structure of the model, but they present only mobility data for

Wuhan and Mainland China without a detailed analysis. Sanche et al (Emerging Infectious Diseases 2020) present an aggregated description of pre-LNY travel from Wuhan to other provinces in China, but again focus mainly on the use of Baidu data as an input into their COVID model. Kraemer et al (Science 2020) use Baidu mobility data in their analysis of mobility and control measures in the COVID-19 epidemic in China, but focus primarily on outflows from Wuhan in their modelling process. As the field of COVID-related modelling in China has been rapidly expanding, are likely to be additional articles and preprints of which I am not aware, but largely the authors appropriately reference the key available published literature in their manuscript.

I am aware of at least one other manuscript that has focused specifically on analysis of Baidu data and travel patterns in China in relation to COVID -- I believe available as a preprint only, posted March 9 on medrxiv.org. The manuscript is by Lai et al ("Assessing spread risk of Wuhan novel coronavirus within and beyond China, January-April 2020: a travel network-based modelling study", <https://doi.org/10.1101/2020.02.04.20020479>.this). This manuscript used historical data from Baidu, however (2013-2015) and did not have access to the 2020 data.

Overall, this manuscript is novel in its focused and detailed analysis of travel patterns using the highly-analyzed Baidu dataset and represents a valuable addition to the literature. I would expect this manuscript to be of interest to readers interested in understanding population movement during the early phases of the epidemic in China. The extent to which the dramatic and country-wide reductions in mobility in China are applicable to other settings is uncertain. As the authors note, extending similar analyses to other locations would help broaden the field's understanding of how travel restrictions influence movement and how this relationship varies by setting.

Overall, this is a thorough and detailed exploration of travel patterns in China during a critical early period in the COVID pandemic. I have a few requests for clarification regarding the methods below, but largely these analyses appear reproducible, and the source code is publicly available online.

There are several areas where I feel that additional clarification, description of methods, or sensitivity analyses would be helpful, which are listed below. The manuscript is, by and large, clearly written, and the methods are broadly appropriate to support the manuscript's conclusions.

Major comments:

- The k-means clustering analysis here is interesting (i.e. page 4 of the PDF version of the manuscript) and argues for the importance of travel to spatially distance areas (including large population centers) early in the epidemic course. I have several questions about this analysis.
 - o Figure 1, panel B: why show the detections of first cases relative to cluster A? This would be easier to interpret if the X-axis showed calendar date and all four clusters were displayed rather than "Earlier" and "Later" than cluster A. This would also allow visual interpretation of the overlaps between the probability densities.
 - o The text states that "Prefectures in cluster D confirmed their first cases 1.08 days earlier [than cluster A]". This is not a very large difference – and the probability densities seem to substantially overlap in Fig 1B. I would recommend that this estimate (1.08 days) be accompanied by a measure of uncertainty, i.e. a 95% uncertainty interval.
 - o The statement that "Cluster A, Cluster B and C detected their first COVID-19 cases at approximately the same time" but cluster D was earlier doesn't seem to be supported by Figure 1b, unless I am reading it incorrectly (the means of the probability distributions are rather evenly distributed, rather than A/B/C being similar with D markedly earlier).
 - o The "detection of first case" analysis attempts to adjust for surveillance intensity per the caption in Figure 1, but reading the methods this is really just an adjustment for population size, which is correlated with group membership (Fig 1c). A few questions about this adjustment:
 - ♣ Could the assumption that of population size is a good proxy for surveillance intensity be validated in some way (e.g. by examining the relationship between tests per capita and population size, if such data exist?). If I understand the methodology correctly, this regression assumes that the relationship between population size and time to first detection (after accounting for travel cluster) is linear. Does the data support this?
 - ♣ Do the key findings (i.e. cluster B, then C with earlier detection of first cases) hold when not adjusted for surveillance intensity (population)? I would presume so, given that the C/D cities tend to be larger, so the unadjusted results may therefore be more dramatic than what is shown – though this would be good to confirm.

♣ In general, more description of this adjustment and the results of the regression in the supplemental materials would be useful to evaluate this model, i.e. the coefficients on the predictors (cluster membership and travel time).

- The discussion of healthcare availability and pressure is a salient one and the key finding – that pre-LNY travel appears to have shifted many people to places with lower healthcare capacity, resulting in a suboptimal distribution of healthcare “pressure” – is of substantial interest. A few comments and questions:

- o I would favor describing this as an analysis of healthcare “capacity” rather than “availability” or “access”. This analysis really is looking at the baseline number of hospitals, rather than whether or not people were actually able to access care (for logistical, social, etc. reasons) as the pandemic progressed.

- o The calculation of healthcare pressure appears to be: [confirmed COVID-19 cases] / [hospitals per 100,000 residents]. Because the numerator is not population-standardized while the denominator is, this metric is prone to some odd behaviors. For instance, if there are 100 COVID cases and one hospital in town A, which has 100,000 residents, the metric’s value is 100 (100 cases / 1 hospital per 100,000 residents). If there are 1,000 COVID cases and one hospital in town B, which has 10,000 residents, the metric’s value is also 100 (1,000 cases / 10 hospitals per 100,000 residents). Would we really expect these two locations to be experiencing the same healthcare pressure? Town A has 100 cases for its one hospital and 1/1,000 residents are infected. Town B has 1,000 cases for its one hospital, and 1/10 residents are infected. I may be misunderstanding this metric, but if the above is true, it seems problematic. Why not use the much simpler metric of cases per hospital? Or (which would have the same result) population-standardize both the case counts and the hospital capacity? A better proxy for health system capacity might be bed space, given that hospital size may vary dramatically, but I appreciate that this information may not be available.

- o Particularly during the early stages of this epidemic, I would expect that case detection would be likely to vary substantially by location and over time. In fact, this is the premise of the population-based adjustment for timing of case detection: that larger locations are likely to have better capacity to detect cases. Have any attempts been made to adjust for the incompleteness (and differential incompleteness) of reporting? Using uncorrected confirmed cases in a time-varying analysis like this could introduce substantial error as detection capacity changes over time and by location.

Minor comments:

- The finding that the temporary increase in outbound travel from Wuhan just prior to the cordon sanitaire was not unique to Wuhan is a valuable one. I am not sure that I fully understand the metrics used (page 14, eq 3 and eq 4), however. I think this is related to the notation. For $V1_i$, for instance, is the first term the peak daily outbound Baidu mobility index score for unit i in the time period of interest in 2020 divided by the mean daily score for the same location in 2019 (minus 1)? In other words, this is just the relative change from mean 2019 mobility index to peak 2020: $(\text{peak}_{2020} - \text{mean}_{2019})/\text{mean}_{2019}$. If so, I would clarify in the notation of the equations that this is the peak 2020 value in the time window in question. This is currently unclear, as F is defined as a daily metric earlier in the methods section.

- SI Fig 13: Unless I missed it, the derivation of these distance kernel plots is not described in neither the methods nor SI. I would suggest that either this analysis be described in more detail in either the methods section or in the SI / figure caption, as it is otherwise very difficult to interpret.

- SI Fig 15: Community modularity. Is it possible to identify which blue node corresponds to Beijing, Shanghai, and Guangzhou/Shenzhen? This would add some useful information to these interesting plots.

References:

- I believe that reference 6 should be Lai, S., Ruktanonchai, N.W., Zhou, L. et al. Effect of non-pharmaceutical interventions to contain COVID-19 in China. Nature (2020).

<https://doi.org/10.1038/s41586-020-2293-x>. It is possible that there is a manuscript with the same name in the journal mentioned here, but seems more likely that this is an outdated reference (the manuscript appears to have now been accepted for publication in Nature).

- Reference 18: is “People” the correct author name?

Reviewer #2 (Remarks to the Author):

No page numbers so these are the page numbers on the pdf document I received.

Overall comments

I think there is some really interesting analysis underlying what appears in this paper, but it feels very much like this has been thrown together in a rush to get something out that is COVID-relevant and as a result, it fails to do some fundamental basic things that I would expect from a high-quality paper. There is some stuff with real merit in here – the migration dataset from Baidu is fantastic and I think what you have tried to do with it has real potential. Bit of a classic more haste, less speed situation. I would recommend focusing on one of the elements and being a lot clearer with the research objectives and the components of the analysis. Unfortunately for me on this occasion this is a revise and resubmit, but I want to impress that I think there is some great analysis underlying this paper, it's just that the way it is presented needs a lot more work and the individual elements need separating out and shoring up.

Issue 1 – The ordering of the sections is jarring. Rearrange the content of the paper so that the methodology comes before the results section (why on earth does it not at present?). As it is, the reader is presented results without explanation and this is maddening – I've just spent half an hour trying to piece together what is being clustered in Figure 1 and it turns out this information comes later in the paper (after the discussion?!?).

Issue 2 – no clear research aims are articulated in the introduction other than a vague notion that we need to understand the effectiveness of travel cordons and human mobility if we are to better understand the spread of disease in a pandemic. Yes, OK, but what do you really want to find out? Is it how specifically the cordon disrupted regular travel patterns and whether this had some kind of spatial dimension? Is it that flow patterns have interesting relationships with the size and spatial arrangement of other cities in China and if we understand how, for example, people tend to flow from big cities back to smaller towns at this time of year? Are we then able to say something about likely disease spread and propagation under different (more normal) travel regimes? Where does the hospital and network analysis fit into this picture? None of this is very clear at all and a much better job needs to be done at the beginning of the paper to set this up. Reference to the wider literature feels a bit scant at the beginning and a better review might set up a more successful articulation of research aims. More appears later on, but again, refer to my very first point.

Issue 3 – results are presented, but without clear linkage to specified aims or objectives. For example, a section in the results refers to movement between prefectures and cities in different population quartiles and there is reference to volumes of travel, patterns related to distance, and a very difficult to interpret graph (sup fig 14) related to mobility and population size quartiles. There's lots here, but the authors do an inadequate job of both interpreting these graphics for the reader and linking the narrative back to so pre-specified objectives. The conclusion on p7: "Therefore, medium sized locations could play a key role in limiting the spread of COVID-19 to prefectures with fewer residents" feels a little like some straws are being clutched at - if they could, then how? It made me think this could well be an urban hierarchy thing. Read David Plane's paper on migration up and down the urban hierarchy - <https://www.pnas.org/content/102/43/15313> - movements of people have been shown to go up and down the hierarchy with larger steps between places that are closer in the hierarchy to each other than those that are further apart. I think the patterns that you are unearthing here – which are very interesting, don't get me wrong – are demonstrating this phenomenon and conclusions like this need to be a little grounded a little more in this sort of theory.

Issue 4 – having read through the paper, I am still none the wiser about what you have actually clustered. Talk of trajectories is confusing as I was immediately thinking of directional trajectories and I could find no direction. I think, now, after going over the paper a couple of times, that the trajectories are temporal ones, but I am still not 100% sure. This could all be cleared up very easily with a section that defines the variables more clearly with a better notation (see some of my suggestions below).

Issue 5 – it feels a bit of a hodgepodge of techniques and methods – which are all pretty cool, I'm sure, but none feel like they are given the attention they deserve in the context of a well-thought-out research narrative. The access to health care bit – yes, very interesting, but it's just stuck in there and doesn't really seem to fit in with the migration stuff. I think the migration bit, the network analysis bit and the hospitals bit could probably all be separated out into different papers and if given the due care and attention required – all make very interesting papers in their own right – as I've already said, there's some good stuff in here. I feel a bit like Greg Wallace on Masterchef feeling a bit sad when the contestant throws too many great ingredients into the dish and it sort of doesn't work.

Some specific comments as the paper is read in a linear fashion – i.e. things I find as I come to them:

P2 – line 13 – 'largest annual human migration' – is this migration or merely seasonal visitation? How long to people spend at their new location after migrating? Week, 2 weeks? Longer? Would also help to clarify the main purpose of this movement – is it to visit family back home? Are these workers? Students? Both? Similar to the Christmas movements in the UK or Thanks Giving in the US? Or are there alternative cultural, spiritual or other motivations? Yes, it might be mentioned when following the link, but a quick note would help very much here.

P3 – how are you determining out-bound travel? Is there some sort of distance threshold or change of region or something that you are using to count out-migrants? Need some clarity here.

P4 – when you refer to a 'trajectory' (the object being clustered) what does this mean? Trajectory would normally imply some kind of direction and distance dimension, but is it not clear how these are incorporated into the objects being clustered

P4 – how are you defining 'neighbours'? Are these all other prefectures in China that are not Wuhan? Or just those we would normally consider as 'neighbours'? e.g. contiguous zones or those within a certain distance. It would be hard to consider Beijing a neighbour, for example. If not neighbours, then 'other prefectures' is fine, but the distinction needs to be clarified.

P4 – I am struggling to see much difference between cluster A and B from figure 1 – how is the cluster defined? This goes back to the earlier point about clearly defining the objects that are being clustered here. Supplementary material – the values in the matrices are what? The values appear to correspond to numbers in the supplied data matrices on github, but the y axis on each graph is not labelled.

OK, you are making me work hard here and I'm piecing together the information slowly – Fig 1 now refers to a 'normalised migration index' and the legend explains that this is 'normalised by the total of each flow'. But I have no idea whether this is total outflows, total inflows or both? In notational form is M is the migration flow between origin i and destination j , the normalised flow could effectively be either $M_{ij}/(\sum_j M_{ij})$ or $M_{ij}/(\sum_i M_{ij})$ or $M_{ij}/(\sum_i \sum_j M_{ij})$ but this only assumes a single temporal dimension.

Right, I have read on a little further and the methods section, bizarrely, comes after the results section. This is very odd – please rearrange this so that your future readers do not have the same

period of head scratching as I did!

P4 – “Compared to Cluster A, Cluster B and C detected their first COVID-19 cases at approximately the same time (Fig 1d). Prefectures in cluster D confirmed their first cases 1.08 days earlier (Figure 1d).” – I think you mean 1b?

P7 – Healthcare availability and migration. Interesting on one level and it kind of makes sense in the context of larger cities having better hospital facilities and most LNY movements moving down the urban hierarchy (one assumes students and workers going back to villages and smaller towns to visit family). This is presented without much further comment, however. Did this lead to an excess of deaths or was it significant in any other way, or do we simply not know? It feels like a bit of a bolt on and again, suffers from the lack of clear research objectives.

P8/9 and Fig 4, reference to ‘modularity’ but no definition – yes a link is given, but again, it would help the reader if this were briefly explained. I think there is something interesting going on with the lead up to LNY and the Wuhan cordon in the community patterns, but again, I am not sure what I am to be taking away from this.

P10 – “We found a limited relationship between spatial proximity and epidemic spread where larger, distant populations detected their first COVID-19 cases earlier than smaller locations that are closer to Wuhan.” – this is interesting and suggests to me that the standard gravity models of migration are probably holding quite well here. Given that the authors are in possession of all of the ingredients to fit a gravity model of migration (population data, origin/destination migration matrices, distance matrices), I would recommend that if this were carried out, they would be able to comment more effectively on the deviations from expected flows. This could be done in using a simple Poisson or negative binomial regression model – in R, something like: `migration_model <- glm(Tij_flow ~ log(origin_pop)+log(destination_pop)+log(dist_ij), family = poisson(link = "log"), data =`

`some_pairwise_rearrangement_of_china_prf_connectivity_0101_0301_plus_the_orig_dest_and_distance_variables)` – it would then be possible to compare fitted values with observed flows and comment on whether the spread of the virus from Wuhan actually did something that we would expect, given the things we know normally influence migration flows, or did it do something quite different?

P13 – There’s some neat data wrangling here, but it could all be specified a little more clearly. I’m not sure equation 2 is quite correct. The ‘overall migration index’

The volume of migration, T_{ij} , between origin, i and destination, j can be calculated:

$$T_{ij} = O_i * P_{ij,O} = D_j * P_{ij,D}$$

Where the overall Baidu Migration Index outbound from origin, O_i and inbound to destination, D_j for $N=366$ prefectures:

$$O_i = (\sum_j^{N} T_{ij}) / T$$

$$D_j = (\sum_i^{N} T_{ij}) / T$$

And two alternative sets of probabilities can be generated:

$$P_{ij,O} = T_{ij} / O_i$$

$$P_{ij,D} = T_{ij} / D_j$$

It is assumed that any time dimension, t , is implicit here with each of these calculated for each of the $t = 1...61$ days in the dataset.

P13/14. Discussion of trajectories – I am still struggling to understand what this trajectory is? Please explain it with reference to the sort of notation I describe above. I think a trajectory is a T_{ij} flow? But then there is talk of an 'outflow trajectory' which can only be O_i as described above? However, I could contest that this is a trajectory as it has no directional component to it (as implied by the term). Again, just not clear and it needs clarifying. Although I am thinking as I go here, and now I am thinking that the trajectory has a temporal rather than directional dimension. This makes more sense, but I should not have to be guessing this.

P14 – following the notes above, please rethink the notation in eq.3 & 4 to incorporate time more effectively.

P15 I'm not sure what the Mann-Whitney U test is telling me? What am I supposed to take away from the list of p-values at the top of p15?

Reviewer #1

Reviewer 1, Remarks to the Author	
I appreciate the thorough and thoughtful responses from the authors. I have a few minor additional comments only.	
Response	We thank the reviewer for this assessment of our manuscript, and for their valuable suggestions.

Reviewer 1, Novelty and Interest	
• I appreciate the additions to the introduction and references. No additional comments.	
Response	We thank the reviewer for this assessment of our changes.

Reviewer 1, Major 1	
• The description of the linear model for surveillance intensity is much more clearly specified now and I appreciate the additional clarification regarding effect sizes and calendar dates.• The revised Fig 1b is much clearer and easier to interpret.• The revised Supp fig 6 figure and revisions to the text more clearly demonstrate the uncertainty regarding the effect size of the detection of the first cases by cluster membership.• I am still a little confused by the new statement in the Results section that “Among the four clusters identified, a high average population was associated with earlier COVID-19 detection (p-value = 0.00002). This association persisted after adjusting for surveillance bias”. This seems circular: I thought that population was the proxy used for surveillance intensity. This statement then seems to suggest that “there is an association between population and earlier COVID-19 detection that persists after adjusting for population.” Perhaps there is a typo here – does this mean to say that there is an association between cluster membership and earlier detection that persists when adjusting for surveillance bias (population as proxy)?	
Response	We are glad that we have clarified the description of the linear model and its interpretation in the figures in the main text and supplement. We agree with the reviewer that the previous text incorrectly stated the relationship between average population and the time of COVID-19 case detection. We provide the following revision on the top of page 5: • Among the four clusters identified, cluster membership was associated with COVID-19 detection timing (p-value = 0.0004).

	Members of cluster D tend to have earlier COVID-19 detection. Such association persisted after adjusting for surveillance bias (p-value = 0.00002, see also Supplemental Figure 6).
--	---

Reviewer 1, Major 2	
I appreciate the expanded discussion in the methods section and Fig 1b revisions as above, as well as the citations. No additional questions or comments regarding this.	
Response	We thank the reviewer for this assessment of our changes.

Reviewer 1, Major 3	
I appreciate the terminology modifications and switch from “access” to “capacity”. In this hypothetical example, the argument that town A is under more healthcare “pressure” and therefore is more susceptible to being overwhelmed by a pandemic than town B is an interesting one. I still think that the analysis that does not standardize by population is more compelling and easy to interpret, using simply “cases per hospital” rather than “cases per (hospitals per 100k residents)”. Regardless, I am satisfied by the justification provided here and particularly appreciate the additional analysis demonstrating that interpretation of the results is largely unchanged regardless of choice of the precise metric.	
Response	We thank the reviewer for this comment and are glad that we have clarified the analysis by including a second metric, “cases per hospital”. We agree that it is helpful to demonstrate to readers that the results are stable regardless of the metric used.

Reviewer 1, Minor 3	
I think that the old SI fig 15 is now SI fig 17. Based on the response, it sounds like the authors had planned to add a color legend to clarify colors, but I don’t see this in the current SI draft – bringing to the authors’ attention if they had intended the addition.	
Response	We believe that the reference to Supplemental Figure 15 is correct in the

	main text, but have added a color legend to Supplemental Figures 17-18, thank you for including this.
--	---

Reviewer #1 response to Reviewer #2

Reviewer 1 to Reviewer 2, Major 1	
Agree with Reviewer #2 that it is sometimes difficult to follow the narrative in methods-heavy papers with the Nature family style of methods at the end. With that said, I believe that the authors have done well to edit the paper to improve flow and clarity within the results section so that the reader can more easily follow the narrative and understand, at least at a high level, what is being done before referring to the methods. Edits made to figures for simplification and captions for clarity have also helped. Figure 1 is easier to interpret without having yet read the methods. In my opinion, the authors have satisfactorily addressed this comment.	
Response	Thank you for this comment, we are glad to have improved the clarity of the work for readers.

Reviewer 1 to Reviewer 2, Major 2	
Edits made to the introduction help better frame and contextualize the manuscript's aims and objectives. There are still a variety of techniques and aims, but the manuscript now better ties them together. In my opinion, the authors have satisfactorily addressed this comment.	
Response	Thank you for this comment, we are glad that the connections between the methods used in the work is now more clear.

Reviewer 1 to Reviewer 2, Major 3	
--

The edits made by the authors clarify their point in the review, i.e. that they are using the movement before LNY as a proxy for long-term migration patterns and a baseline for comparison to the patterns observed during LNY.

I am not an expert in theories of migration, so my ability to evaluate this particular critique is limited. The authors do appear to cite the literature referenced by reviewer #2 appropriately from my review of the cited articles and include a source specific to the context of China. This seems to constitute “grounding [the observations] a little more in this sort of [migration] theory” as requested by the reviewer.

In addition, the revised text does improve the interpretation of the complex analyses and graphs for the reader, particularly as relates to the potential policy implications.

In my opinion, the authors have satisfactorily addressed this comment.

Response

Thank you for this comment, the connection to existing theories of migration was valuable to link our findings to an existing theoretical foundation.

Reviewer 1 to Reviewer 2, Major 4

The clarifications provided by the authors are an improvement over the previous language. The k-means clustering in the beginning of the sentences mentioned here appropriately describes the methodology – and I appreciate that from a statistical sense these are indeed clusters of temporal patterns -- but I think that describing “four clusters” as the outputs of the analysis is the part that is causing the confusion. I think that part of the confusion here is that many readers (and perhaps reviewer #2) might assume that “clustering” is spatial, particularly in a paper that is highly focused on spatial patterning in many of its analyses. One might mis-interpret “temporal clusters” to indicate time-varying clusters in space (for instance, if someone described “temporal clusters” of COVID disease to me, I would probably assume that they’re doing a spatiotemporal analysis). I wonder if it would be better to say that using k-means clustering, the analysis identified four general categories / archetypes / etc. of travel patterns from Wuhan, or some other phrasing that avoids the “cluster” terminology to describe the groupings that the analysis produces.. This is a relatively fine point and could probably be handled in the editing process.

Response

We have updated the text to describe “four patterns” rather than “four clusters.” We have also replaced the use of “temporal trajectory” with “time series” at all occurrences.

Reviewer 1 to Reviewer 2, Major 5	
The edits help to tie together the separate analyses and results, particularly better linking in the healthcare capacity analysis. The one part of the analysis that is still a bit on its own is the “Changes in overall travel network structure”. This section could use a little more of the treatment that other sections received – in particular, setting up the motivation for the analysis and doing a bit more to help the reader interpret the results. Again a relatively minor point and I think that the authors have largely addressed Reviewer #2’s comment here.	
Response	We agree that the section “Changes in overall travel network structure” could improve from more explanation of its position in the broader motivations for this work. We have added the following in an effort to clarify the motivation for this portion of the study:  • A lower Modularity value (Q) indicates weaker connections between prefectures within a community, or higher volumes of travel between different communities, rather than within the same communities. The low values of Q preceding LNY indicate a high volume of travel between communities, with increased interconnection of the movement network.

Reviewer 1 to Reviewer 2, Minor 1	
Agree that the switch from “migration” to “movement” throughout the manuscript is beneficial. The other edits made, in my opinion, adequately address Reviewer #2’s comments.	
Response	Thank you for this comment, we are glad that the change is beneficial.

Reviewer 1 to Reviewer 2, Minor 2	
In my opinion, the authors have satisfactorily addressed this comment with this edit	
Response	Thank you for this comment.

Reviewer 1 to Reviewer 2, Minor 3	
See my comment above regarding the “clustering of trajectories” and the potential confusion around the word “cluster” to readers, although from a statistical perspective the term is appropriate.	
Response	We have changed the text from “clustering of trajectories” to “clustering of time series” in an effort to reduce confusion.

Reviewer 1 to Reviewer 2, Minor 4	
In my opinion, the authors have satisfactorily addressed this comment with this edit	
Response	Thank you for this comment.

Reviewer 1 to Reviewer 2, Minor 5	
Appreciate the edits, which are helpful. I think that some of the language in this response (avoiding the term “trajectory” as noted in prior comments, but emphasizing that the k-means clustering is intended to categorize similarly shaped time series into four groupings) may be helpful to add to the manuscript to address some of the confusion that reviewer #2 expresses regarding the “cluster” / “trajectory” terminology.	
Response	We have removed references to “trajectories” and replaced them with “time series”. We hope that this more clearly identifies that we are clustering temporal, not spatial, patterns.

Reviewer 1 to Reviewer 2, Minor 6	
The additional paragraph and equation are helpful. If I am understanding correctly, it seems like origin (i) is always Wuhan in this example, and it may be helpful to actually clarify in this paragraph that this is the normalized flow, N , between origin (i) – in this case, Wuhan – and destination (j). This is mentioned in the preceding paragraph but it seems like some of reviewer #2’s question was related to the fact that the normalized migration index could be used to generate pairwise comparisons between all combinations of prefectures. Directly specifying that Wuhan is always the origin, if that’s indeed the case, might therefore be helpful. Again a relatively minor point.	
Response	We agree that it could be clarified that in this analysis, the only origin location is Wuhan in the main paper, but the method could be applied to all pairs of prefectures, and is repeated for 5 other prefectures in the Supplement.

	We have updated the text accordingly:  In order to characterise the shape of the outflow trajectories from Wuhan, rather than the magnitude of certain outflows, we calculated the normalised flow, N, between origin (i) and destination (j) prefectures on each day (t), by dividing the outflow measured by the travel index T, by the total movement between the 1st and 23rd January 2020
--	---

Reviewer 1 to Reviewer 2, Minor 7	
If structure is mandated by Nature Communications, which I think is indeed the case, then this comment would be irrelevant	
Response	The arrangement of the manuscript sections complies with the editorial policies of Nature Communications.

Reviewer 1 to Reviewer 2, Minor 8	
Revisions are satisfactory	
Response	Thank you for this comment.

Reviewer 1 to Reviewer 2, Minor 9	
In my opinion, the authors have satisfactorily addressed this comment with these edits and additions.	
Response	Thank you for this comment.

Reviewer 1 to Reviewer 2, Minor 10	
The edits made do clarify what modularity is measuring. Reviewer #2 also notes, however, "I think there is something interesting going on with the lead up to LNY and the Wuhan cordon in the community patterns, but again, I am not sure what I am to be taking away from this." I don't think that the edits address this part of the reviewer's comment – what is the major takeaway from this analysis? As in my response above to Major Comment #5, I think that this section is the one that could use a little more detail re: the motivation for the analysis and a little more hand-holding for the reader in the interpretation of the results.	

Response	We agree that this could be explained more clearly and have added text to describe the relation between Modularity (Q) and the connections between prefectures in different communities (included above).
----------	---

Reviewer 1 to Reviewer 2, Minor 11	
I think that the authors provide a reasonable defense of their methods and choice here. The proposed analysis from reviewer #2 would indeed be interesting if possible, but I think it's reasonable for the authors to say that it's outside the scope of this analysis	
Response	Thank you for this comment.

Reviewer 1 to Reviewer 2, Minor 12	
In my opinion, the authors have satisfactorily addressed this comment with these edits, and the description / notation is clearer now.	
Response	Thank you for this comment.

Reviewer 1 to Reviewer 2, Minor 13	
My same comments above regarding the cluster / trajectory / time series terminology apply here, though I think that the edits made so far do substantially improve the reader's ability to follow the analysis (i.e. removing "trajectory").	
Response	We have altered the wording of "clustering" and "trajectories" as described above.

Reviewer 1 to Reviewer 2, Minor 14	
Explicit addition of (t) subscripts does help notation. While the notation is clear, the text description currently reads "Where $T_{(ij,t)}$ is the volume of mobility from location i to location j on day t, F is the overall Baidu migration index with direction (inbound or outbound) at location i, and $p_{(i,j,outbound)}$ is the proportion of all outbound travel that originated in i and ended in j." It might be more clear to also indicate that F and p both also correspond to day t, for instance: "Where, for a given day t, $T_{(ij,t)}$ is the volume of mobility from location i to location j, F is the overall Baidu migration index with direction (inbound or	

outbound) at location i, and $p_{(i,j, \text{outbound})}$ is the proportion of all outbound travel that originated in i and ended in j.” – or something similar. This is a minor point only and could be addressed in editing stages	
Response	We agree that this addition clarifies the discussion of this equation. The text now reads:  • Where, for a given day, $T_{ij,t}$ is the volume of mobility from location i to location j on day t, F is the overall Baidu migration index with direction (inbound or outbound) at location i, and $p_{ij_outbound}$ is the proportion of all outbound travel that originated in i and ended in j.

Reviewer 1 to Reviewer 2, Minor 15	
The responses and edits to the manuscript do clarify the analysis substantially. The use of a one-tailed Mann-Whitney U makes a relatively strong assumption about the alternative hypothesis (excluding the possibility that healthcare pressures in high healthcare capacity settings would be higher than those in low healthcare capacity settings from the alternative hypothesis). If there were many more confirmed cases in high healthcare capacity settings early in the outbreak, e.g. if the outbreak happened to spread first to places with high healthcare capacities, it seems that it would have been plausible for there to have been higher healthcare pressures (according to the authors’ metric) in the high-capacity compared to the low-capacity settings. The authors should probably provide some additional justification for the use of a one-tailed test – unless I’m missing something that would make it impossible for the high capacity settings to experience more pressure than the low capacity settings -- or use a two-tailed test.	
Assigned to	Hamish, Yang
Response	Thank you for this comment. We decided to conduct two-tail tests as well as one-tailed tests in the other direction. We realised this may not have come across in the manuscript. Thus:  • On page 16, ‘Assessing the Healthcare Capacity and COVID-19 related Healthcare Pressure’, we added the following content to the end:  ○ Results were verified using two-tailed Mann-Whitney U tests, as well as one-tailed Mann-Whitney U tests with the opposite null hypotheses. • In the supplemental material, on page 22, we have added the

following content to the caption:

Assessing healthcare capacity related to COVID-19 using an alternate metric, the number of Grade II and Grade III hospitals in each prefecture, without adjusting for population size ($n_{low} = 154$, $n_{high} = 156$). Similar to figure 3 in the main text, the statistical significance (shown in this figure as shades, i.e., darker color indicates statistical significance) is based on two-tail Mann Whitney U tests ($n_1 = 154$, $n_2 = 156$). The null hypothesis was that the healthcare pressures in settings with fewer hospitals are comparable to that in settings with more hospitals; the alternative hypothesis was that healthcare pressures in settings with fewer hospitals are higher than those in settings with more hospitals. This test was repeated for each week from week three to nine (i.e., starting on 15 Jan 2020). Results were verified using two-tailed Mann-Whitney U tests, as well as one-tailed Mann-Whitney U tests with the opposite null hypotheses. In this process, we found that in week 9, healthcare pressure in places with more hospitals is higher. Pre-LNY travels may no longer drive COVID-19 related healthcare pressure later on.

REVIEWERS' COMMENTS:

Reviewer #1 (Remarks to the Author):

I appreciate the thorough and thoughtful responses from the authors. I have a few minor additional comments only:

Reviewer #1 Novelty and Interest:

- I appreciate the additions to the introduction and references. No additional comments.

Reviewer #1, Major #1:

- The description of the linear model for surveillance intensity is much more clearly specified now and I appreciate the additional clarification regarding effect sizes and calendar dates.
- The revised Fig 1b is much clearer and easier to interpret.
- The revised Supp fig 6 figure and revisions to the text more clearly demonstrate the uncertainty regarding the effect size of the detection of the first cases by cluster membership.
- I am still a little confused by the new statement in the Results section that "Among the four clusters identified, a high average population was associated with earlier COVID-19 detection (p -value = 0.00002). This association persisted after adjusting for surveillance bias". This seems circular: I thought that population was the proxy used for surveillance intensity. This statement then seems to suggest that "there is an association between population and earlier COVID-19 detection that persists after adjusting for population." Perhaps there is a typo here – does this mean to say that there is an association between cluster membership and earlier detection that persists when adjusting for surveillance bias (population as proxy)?

Reviewer #1, Major #2:

- I appreciate the expanded discussion in the methods section and Fig 1b revisions as above, as well as the citations. No additional questions or comments regarding this.

Reviewer #1, Major #3:

- I appreciate the terminology modifications and switch from "access" to "capacity".
- In this hypothetical example, the argument that town A is under more healthcare "pressure" and therefore is more susceptible to being overwhelmed by a pandemic than town B is an interesting one. I still think that the analysis that does not standardize by population is more compelling and easy to interpret, using simply "cases per hospital" rather than "cases per (hospitals per 100k residents)". Regardless, I am satisfied by the justification provided here and particularly appreciate the additional analysis demonstrating that interpretation of the results is largely unchanged regardless of choice of the precise metric.

For the minor comments, I appreciate the revisions and clarifications.

- Reviewer #1, Minor #3: I think that the old SI fig 15 is now SI fig 17. Based on the response, it sounds like the authors had planned to add a color legend to clarify colors, but I don't see this in the current SI draft – bringing to the authors' attention if they had intended the addition.

Assessment by Reviewer 1 of response to comments made by Reviewer 2:

Reviewer #2 Comments:

Major comment #1:

Agree with Reviewer #2 that it is sometimes difficult to follow the narrative in methods-heavy papers with the Nature family style of methods at the end. With that said, I believe that the authors have done well to edit the paper to improve flow and clarity within the results section so that the reader can more easily follow the narrative and understand, at least at a high level, what

is being done before referring to the methods.

Edits made to figures for simplification and captions for clarity have also helped. Figure 1 is easier to interpret without having yet read the methods.

In my opinion, the authors have satisfactorily addressed this comment.

Major comment #2:

Edits made to the introduction help better frame and contextualize the manuscript's aims and objectives. There are still a variety of techniques and aims, but the manuscript now better ties them together.

In my opinion, the authors have satisfactorily addressed this comment.

Major comment #3:

The edits made by the authors clarify their point in the review, i.e. that they are using the movement before LNY as a proxy for long-term migration patterns and a baseline for comparison to the patterns observed during LNY.

I am not an expert in theories of migration, so my ability to evaluate this particular critique is limited. The authors do appear to cite the literature referenced by reviewer #2 appropriately from my review of the cited articles and include a source specific to the context of China. This seems to constitute "grounding [the observations] a little more in this sort of [migration] theory" as requested by the reviewer.

In addition, the revised text does improve the interpretation of the complex analyses and graphs for the reader, particularly as relates to the potential policy implications.

In my opinion, the authors have satisfactorily addressed this comment.

Major comment #4:

The clarifications provided by the authors are an improvement over the previous language.

The k-means clustering in the beginning of the sentences mentioned here appropriately describes the methodology – and I appreciate that from a statistical sense these are indeed clusters of temporal patterns -- but I think that describing "four clusters" as the outputs of the analysis is the part that is causing the confusion. I think that part of the confusion here is that many readers (and perhaps reviewer #2) might assume that "clustering" is spatial, particularly in a paper that is highly focused on spatial patterning in many of its analyses. One might mis-interpret "temporal clusters" to indicate time-varying clusters in space (for instance, if someone described "temporal clusters" of COVID disease to me, I would probably assume that they're doing a spatiotemporal analysis). I wonder if it would be better to say that using k-means clustering, the analysis identified four general categories / archetypes / etc. of travel patterns from Wuhan, or some other phrasing that avoids the "cluster" terminology to describe the groupings that the analysis produces.. This is a relatively fine point and could probably be handled in the editing process.

Major comment #5:

The edits help to tie together the separate analyses and results, particularly better linking in the healthcare capacity analysis.

The one part of the analysis that is still a bit on its own is the "Changes in overall travel network structure". This section could use a little more of the treatment that other sections received – in particular, setting up the motivation for the analysis and doing a bit more to help the reader interpret the results. Again a relatively minor point and I think that the authors have largely addressed Reviewer #2's comment here.

Minor comment #1:

Agree that the switch from "migration" to "movement" throughout the manuscript is beneficial. The other edits made, in my opinion, adequately address Reviewer #2's comments.

Minor comment #2:

In my opinion, the authors have satisfactorily addressed this comment with this edit

Minor comment #3:

See my comment above regarding the "clustering of trajectories" and the potential confusion around the word "cluster" to readers, although from a statistical perspective the term is appropriate.

Minor comment #4:

In my opinion, the authors have satisfactorily addressed this comment with this edit

Minor comment #5:

Appreciate the edits, which are helpful. I think that some of the language in this response (avoiding the term "trajectory" as noted in prior comments, but emphasizing that the k-means clustering is intended to categorize similarly shaped time series into four groupings) may be helpful to add to the manuscript to address some of the confusion that reviewer #2 expresses regarding the "cluster" / "trajectory" terminology.

Minor comment #6:

The additional paragraph and equation are helpful. If I am understanding correctly, it seems like origin (i) is always Wuhan in this example, and it may be helpful to actually clarify in this paragraph that this is the normalized flow, N , between origin (i) – in this case, Wuhan – and destination (j). This is mentioned in the preceding paragraph but it seems like some of reviewer #2's question was related to the fact that the normalized migration index could be used to generate pairwise comparisons between all combinations of prefectures. Directly specifying that Wuhan is always the origin, if that's indeed the case, might therefore be helpful. Again a relatively minor point.

Minor comment #7:

If structure is mandated by Nature Communications, which I think is indeed the case, then this comment would be irrelevant

Minor comment #8:

Revisions are satisfactory

Minor comment #9:

In my opinion, the authors have satisfactorily addressed this comment with these edits and additions.

Minor comment #10:

The edits made do clarify what modularity is measuring.

Reviewer #2 also notes, however, "I think there is something interesting going on with the lead up to LNY and the Wuhan cordon in the community patterns, but again, I am not sure what I am to be taking away from this." I don't think that the edits address this part of the reviewer's comment – what is the major takeaway from this analysis? As in my response above to Major Comment #5, I think that this section is the one that could use a little more detail re: the motivation for the analysis and a little more hand-holding for the reader in the interpretation of the results.

Minor comment #11:

I think that the authors provide a reasonable defense of their methods and choice here. The proposed analysis from reviewer #2 would indeed be interesting if possible, but I think it's reasonable for the authors to say that it's outside the scope of this analysis

Minor comment #12:

In my opinion, the authors have satisfactorily addressed this comment with these edits, and the description / notation is clearer now.

Minor comment #13:

My same comments above regarding the cluster / trajectory / time series terminology apply here, though I think that the edits made so far do substantially improve the reader's ability to follow the analysis (i.e. removing "trajectory").

Minor comment #14:

Explicit addition of (t) subscripts does help notation.

While the notation is clear, the text description currently reads "Where $T_{ij,t}$ is the volume of mobility from location i to location j on day t , F is the overall Baidu migration index with direction (inbound or outbound) at location i , and $p_{(i,j,\text{outbound})}$ is the proportion of all outbound travel that originated in i and ended in j ." It might be more clear to also indicate that F and p both also correspond to day t , for instance: "Where, for a given day t , $T_{ij,t}$ is the volume of mobility from location i to location j , F is the overall Baidu migration index with direction (inbound or outbound) at location i , and $p_{(i,j,\text{outbound})}$ is the proportion of all outbound travel that originated in i and ended in j ." – or something similar. This is a minor point only and could be addressed in editing stages

Minor comment #15:

The responses and edits to the manuscript do clarify the analysis substantially.

The use of a one-tailed Mann-Whitney U makes a relatively strong assumption about the alternative hypothesis (excluding the possibility that healthcare pressures in high healthcare capacity settings would be higher than those in low healthcare capacity settings from the alternative hypothesis). If there were many more confirmed cases in high healthcare capacity settings early in the outbreak, e.g. if the outbreak happened to spread first to places with high healthcare capacities, it seems that it would have been plausible for there to have been higher healthcare pressures (according to the authors' metric) in the high-capacity compared to the low-capacity settings. The authors should probably provide some additional justification for the use of a one-tailed test – unless I'm missing something that would make it impossible for the high capacity settings to experience more pressure than the low capacity settings -- or use a two-tailed test.

Response to reviewers

Reviewer #1

Reviewer 1, Key results summary	
In this manuscript, Gibbs et al analyze travel data from Baidu Huiyan to describe travel patterns in China during the emergence of the COVID-19 epidemic in January and February 2020. Changes in travel flows and travel network structure are analyzed in the context of key events over the relative time period, such as the Lunar New Year and the implementation of a cordon sanitaire. The key analyses and findings of this paper include: • An analysis of temporal patterns of travel between Wuhan and other prefectures in China. In comparison to 2019, the authors identify a 2020 peak in outbound travel in early January that is difficult to definitively explain, as well as a higher peak just preceding the start of the lunar new year. The authors do not find evidence for an anecdotally-reported, anomalous, Wuhan-specific increase in outbound travel just prior to or after the announcement of the cordon sanitaire.• A cluster analysis of patterns of outbound travel from Wuhan. The authors use k-means clustering to create four groups of destination prefectures according to temporal trajectories of outbound travel from Wuhan. Combining these groupings with an analysis of timing of first case detection, this analysis suggests that travel to more spatially distant, more populous prefectures was relatively higher in early January compared to just before LNY, and the timing of first case detection was slightly earlier (~1 day) in these destination prefectures. Travel to more spatially proximate prefectures was most common overall, peaked just prior to the LNY, and was associated with slightly later detection of first cases.• An analysis of movement patterns between prefectures grouped by population quartile. Pre-LNY movement indicated increased movement out of highly populated prefectures to other locations, particularly those with lower population sizes. After the cordon sanitaire, overall population flows decreased markedly, but relatively more travel originated from less-populated prefectures.• An analysis of healthcare capacity and population size. During the pre-LNY period, travel from areas with higher to areas with lower healthcare capacity was common. The authors generate a measure of healthcare pressure using confirmed cases and capacity and suggest that pre-LNY travel shifted healthcare pressure to places that had low baseline capacity.• An analysis of the prefecture-level travel network structure and the evolution of modularity over time. This analysis suggests that travel flow and connectivity between prefectures decreased broadly throughout China following the cordon sanitaire. There was a brief restructuring immediately following the cordon sanitaire, with Wuhan assuming increased importance in the network – perhaps due to outbreak response activities, the authors suggest – but a rapid reversion to the pre-LNY state and no persistent substantial restructuring of the travel network.	
Response	We thank the reviewer for these positive comments on the manuscript,

	and for identifying the key areas of novelty.
--	---

Reviewer 1 Novelty and interest	
As the authors appropriately note in the introduction, Baidu mobility data has been used from in several other analyses of COVID in China. For example, Lai et al. (Nature 2020) use Baidu mobility data as an input into a travel network-based SEIR model simulating COVID spread; they briefly describe the daily outflow from 340 cities within mainland China but their main focus is in modeling of the epidemic. They do not assess travel patterns in early January 2020. Chinazzi et al (Science 2020) similarly use Baidu mobility data as an input into an individual-based stochastic spatial epidemic model. This analysis uses Baidu mobility indices at the province level to parameterize the metapopulation structure of the model, but they present only mobility data for Wuhan and Mainland China without a detailed analysis. Sanche et al (Emerging Infectious Diseases 2020) present an aggregated description of pre-LNY travel from Wuhan to other provinces in China, but again focus mainly on the use of Baidu data as an input into their COVID model. Kraemer et al (Science 2020) use Baidu mobility data in their analysis of mobility and control measures in the COVID-19 epidemic in China, but focus primarily on outflows from Wuhan in their modelling process. As the field of COVID-related modelling in China has been rapidly expanding, are likely to be additional articles and preprints of which I am not aware, but largely the authors appropriately reference the key available published literature in their manuscript. I am aware of at least one other manuscript that has focused specifically on analysis of Baidu data and travel patterns in China in relation to COVID -- I believe available as a preprint only, posted March 9 on medrxiv.org. The manuscript is by Lai et al ("Assessing spread risk of Wuhan novel coronavirus within and beyond China, January-April 2020: a travel network-based modelling study", https://doi.org/10.1101/2020.02.04.20020479). This manuscript used historical data from Baidu, however (2013-2015) and did not have access to the 2020 data. Overall, this manuscript is novel in its focused and detailed analysis of travel patterns using the highly-analyzed Baidu dataset and represents a valuable addition to the literature. I would expect this manuscript to be of interest to readers interested in understanding population movement during the early phases of the epidemic in China. The extent to which the dramatic and country-wide reductions in mobility in China are applicable to other settings is uncertain. As the authors note, extending similar analyses to other locations would help broaden the field's understanding of how travel restrictions influence movement and how this relationship varies by setting. Overall, this is a thorough and detailed exploration of travel patterns in China during a critical early period in the COVID pandemic. I have a few requests for clarification regarding the methods below, but largely these analyses appear reproduceable, and the source code is publicly available online. There are several areas where I feel that additional clarification, description of methods, or sensitivity analyses would be helpful, which are listed below. The manuscript is, by and large, clearly written, and the methods are broadly appropriate to support the manuscript's conclusions.	
Response	We thank the reviewer for these positive comments. We had included some of

	these references already, but we have added the others in an expanded section of the introduction, and thank the reviewer for this excellent summary. We have added the following to the introduction: Several studies have focused on assessing the effectiveness of the cordon sanitaire in Wuhan and other domestic travel restrictions in China in the context of COVID-19 control⁶⁻⁸. As other affected regions worldwide begin implementing similar travel restrictions⁹, it is critical to understand human mobility patterns during the initial phase of the COVID-19 pandemic and their potential implications for other countries. And: Previous analyses of Baidu movement data have used mobility data in transmission models^{6,11}, and others have examined the changes in patterns around Wuhan⁷. A key unknown is to what extent the observed travel patterns in Wuhan and the rest of China were part of regular seasonal movements or were responses to the emerging epidemic or interventions against it, including the cordon sanitaire.
--	---

Reviewer 1, Major 1	
 • The k-means clustering analysis here is interesting (i.e. page 4 of the PDF version of the manuscript) and argues for the importance of travel to spatially distance areas (including large population centers) early in the epidemic course. I have several questions about this analysis.  o Figure 1, panel B: why show the detections of first cases relative to cluster A? This would be easier to interpret if the X-axis showed calendar date and all four clusters were displayed rather than “Earlier” and “Later” than cluster A. This would also allow visual interpretation of the overlaps between the probability densities. o The text states that “Prefectures in cluster D confirmed their first cases 1.08 days earlier [than cluster A]”. This is not a very large difference – and the probability densities seem to substantially overlap in Fig 1B. I would recommend that this estimate (1.08 days) be accompanied by a measure of uncertainty, i.e. a 95% uncertainty interval. o The statement that “Cluster A, Cluster B and C detected their first COVID-19 cases at approximately the same time” but cluster D was earlier doesn’t seem to be supported by Figure 1b, unless I am reading it incorrectly (the means of the probability distributions are rather evenly distributed, rather than A/B/C being similar with D markedly earlier). 	
Response	Thank you for this comment. To adjust for potential bias due to surveillance intensity, we used a linear model to characterise the relationship between cluster membership and detection time

of the first case:

$$\text{Detection Date} \sim \beta_0 + \beta_1 * \text{pop} + \beta_2 * (\text{Cluster Membership})$$

where population is used as a proxy for surveillance intensity, and cluster membership is included as a nominal categorical variable. The implicit assumption here is that places with larger population commit more surveillance efforts. The impact of cluster membership on first-case-detection time is essentially the effect size of the Cluster Membership variable and thus does not have a calendar date attached to it. Cluster A is the reference baseline because it is the reference nominal categorical level in the regression model. It is difficult to translate these effect sizes into specific calendar dates without imposing assumptions about population size.

That being said, we agree that the current presentation of these results is slightly confusing to readers. We have made the following revisions to the manuscript:

- The adjusted timing of first detection shown in Figure 1b has been replaced by the distribution of first detection dates, presented as boxplots. This is an intuitive representation of the relationship between first-case-detection and cluster membership:

- The original Figure 1b has been moved to the supplemental material Figure 6. The original density plot has been revised to a boxplot showing interquartile range as well as the 95% uncertainty range.

Supplemental Figure 6. Box plot of the effect size of the detection of first cases by cluster membership.

The effect size of detection of first case relative to cluster A for clusters B, C, and D. Boxplots show interquartile range and 95% confidence interval (B: -2.0167–2.9814, C: -3.5778–2.3481, D: -4.0883–1.9342). For linear model of effect size given cluster membership normalised by population: $R^2 = 0.1769$ (pictured). For linear model of effect size given cluster membership normalised by log of population: $R^2 = 0.1654$.

- In the Methods section: “Trajectory Analysis and Surge Evaluation” the following paragraph has been removed:

The association between the arrival of the first case and travel clusters were explored using a linear model with two independent variables, travel cluster and population. Population sizes of prefectures were used as a proxy of surveillance intensity, i.e. we assumed that more public health surveillance was conducted in larger cities. The coefficients and the standard errors of travel clusters were then compared.

- We then expanded the Methods section: “The Relationship between First Case Detection and Cluster Membership”, to give more details on the statistical adjustment:

We explored the association between average population size and first case detection for prefectures in each cluster. There is potential confounding due to surveillance bias such that larger prefectures may detect COVID-19 cases earlier due to better public health infrastructure resulting in earlier and greater use of diagnostic tests. However, there is no intuitive indicator that can capture surveillance efforts, and therefore we used population size as a proxy for surveillance effort. The implicit assumption is that places with larger populations are more equipped for detecting COVID-19, which is supported because early testing capacity relied on biosafety level 2+ laboratories, which are only found in large hospitals and universities³⁹.

We adjust for this potential confounding effect using a linear regression model:

$$\text{Detection Date} \sim \beta_0 + \beta_1 * \text{pop} + \beta_2 * (\text{Cluster Membership}) \text{ (eq. 6)}$$

where pop represents the prefecture level population size as of 2018 and (Cluster Membership) is a nominal categorical variable with levels A through D.

- In the Results section, “Human movement surrounding the epicentre – Wuhan, Hubei” subsection, we clarify the relationship between first-case

	detection and cluster membership with the following text: Among the four clusters identified, a high average population was associated with earlier COVID-19 detection (p-value = 0.00002). This association persisted after adjusting for surveillance bias (Supplemental Figure 6). Cluster membership was also associated with prefecture level population sizes (Figure 1c). The Cluster D includes large population centres (e.g. Beijing, Shanghai, Guangzhou, and Shenzhen) (Figure 1d).
--	--

Reviewer 1, Major 2	
o The “detection of first case” analysis attempts to adjust for surveillance intensity per the caption in Figure 1, but reading the methods this is really just an adjustment for population size, which is correlated with group membership (Fig 1c). A few questions about this adjustment: § Could the assumption that of population size is a good proxy for surveillance intensity be validated in some way (e.g. by examining the relationship between tests per capita and population size, if such data exist?). If I understand the methodology correctly, this regression assumes that the relationship between population size and time to first detection (after accounting for travel cluster) is linear. Does the data support this? § Do the key findings (i.e. cluster B, then C with earlier detection of first cases) hold when not adjusted for surveillance intensity (population)? I would presume so, given that the C/D cities tend to be larger, so the unadjusted results may therefore be more dramatic than what is shown – though this would be good to confirm. § In general, more description of this adjustment and the results of the regression in the supplemental materials would be useful to evaluate this model, i.e. the coefficients on the predictors (cluster membership and travel time).	
Response	Thank you for this comment. We agree that the original term used, “surveillance intensity”, may not be easy for readers to interpret. We were trying to describe bias that may exist in the capacity to detect infections. “Surveillance intensity” is not necessarily accurate because it may also reflect emergency response efforts. At the beginning of the outbreak, the capacity to detect infections (through testing swabs) largely relied on the number and capacity of biosafety level 2+ laboratories, most of which are found in large hospitals and provincial level health departments. Thus, we believe population size can be an appropriate proxy for the capacity to detect infections. To address the points raised in this comment, we provide the following revisions:  • We have altered Figure 1b from the adjusted timing of first detection to

	the raw timing of first detection, presented as boxplots. This is an intuitive representation of the relationship between first case detection and cluster membership.  • The original Figure 1b has been revised to a boxplot showing interquartile range and 95% uncertainty range, now included as Supplemental Figure 6. • We have expanded the methods section to provide more details for the statistical adjustment, and added justification for the use of population as a proxy: The Relationship between First Case Detection and Cluster Membership We explored the association between average population size and first case detection for prefectures in each cluster. There is potential confounding due to surveillance bias such that larger prefectures may detect COVID-19 cases earlier due to better public health infrastructure resulting in earlier and greater use of diagnostic tests. However, there is no intuitive indicator that can capture surveillance efforts, and therefore we used population size as a proxy for surveillance effort. The implicit assumption is that places with larger populations are more equipped for detecting COVID-19, which is supported because early testing capacity relied on biosafety level 2+ laboratories, which are only found in large hospitals and universities³⁹. We adjust for this potential confounding effect using a linear regression model: $\text{Detection Date} \sim \beta_0 + \beta_1 * \text{pop} + \beta_2 * (\text{Cluster Membership}) \text{ (eq. 6)}$ where pop represents the prefecture level population size as of 2018 and (Cluster Membership) is a nominal unordered categorical variable with levels A through D.  • We have added a citation which describes how early detection capacity heavily relies on laboratories in large hospitals and universities, justifying using population sizes as a proxy. 39. Cui, X. From 200 to 10,000 a day - how did Wuhan's PCR testing capacity for COVID-19 increased 50 folds? China Times (2020).
--	--

Reviewer 1, Major 3
 • The discussion of healthcare availability and pressure is a salient one and the key finding – that pre-LNY travel appears to have shifted many people to places with lower healthcare capacity, resulting in a suboptimal distribution of healthcare “pressure” – is of substantial

interest. A few comments and questions:

o I would favor describing this as an analysis of healthcare “capacity” rather than “availability” or “access”. This analysis really is looking at the baseline number of hospitals, rather than whether or not people were actually able to access care (for logistical, social, etc. reasons) as the pandemic progressed.

o The calculation of healthcare pressure appears to be: [confirmed COVID-19 cases] / [hospitals per 100,000 residents]. Because the numerator is not population-standardized while the denominator is, this metric is prone to some odd behaviors. For instance, if there are 100 COVID cases and one hospital in town A, which has 100,000 residents, the metric’s value is 100 (100 cases / 1 hospital per 100,000 residents). If there are 1,000 COVID cases and one hospital in town B, which has 10,000 residents, the metric’s value is also 100 (1,000 cases / 10 hospitals per 100,000 residents). Would we really expect these two locations to be experiencing the same healthcare pressure? Town A has 100 cases for its one hospital and 1/1,000 residents are infected. Town B has 1,000 cases for its one hospital, and 1/10 residents are infected. I may be misunderstanding this metric, but if the above is true, it seems problematic. Why not use the much simpler metric of cases per hospital? Or (which would have the same result) population-standardize both the case counts and the hospital capacity? A better proxy for health system capacity might be bed space, given that hospital size may vary dramatically, but I appreciate that this information may not be available.

o Particularly during the early stages of this epidemic, I would expect that case detection would be likely to vary substantially by location and over time. In fact, this is the premise of the population-based adjustment for timing of case detection: that larger locations are likely to have better capacity to detect cases. Have any attempts been made to adjust for the incompleteness (and differential incompleteness) of reporting? Using uncorrected confirmed cases in a time-varying analysis like this could introduce substantial error as detection capacity changes over time and by location.

Response

We agree with the reviewer that “healthcare capacity” is the more appropriate term and we have changed all “healthcare access” to “healthcare capacity”. We have made this change throughout the draft, and thus will not reflect on each change here in the response letter.

Regarding the specific methods to calculate the healthcare pressure due to COVID-19, the reviewer provided an example

- Town A has 100 cases, 1 hospital, and 100,000 residents, the metric is $100/(1/100000) = 10^7$
- Town B has 1000 cases, 1 hospital, and 10,000 residents, the metric is $1000/(1/10000) = 10^7$

With the current equation, the two towns do have the same healthcare pressure. The reason is town A has higher year-round medical care pressure – the one hospital needs to service 100,000 residents in town A whereas in town B, the one hospital only need to service 10,000 on a regular basis. Thus, in an outbreak setting, it takes fewer cases for town A to experience the same pressure as town B.

The implicit assumption here is hospital A and B are comparable. In this study, we used only Grade II and III hospitals, which have standardised sizes in China. By definition, Grade II hospitals have 100-499 beds and Grade III hospitals have 500+ beds and are both equipped with ventilators. Therefore, we think the issue of hospital size is minimal (although well noted) and our assumption is defensible. Additionally, information on hospital beds available per 1,000 residents is not publicly available on the prefecture level.

Following the reviewer's suggestion we experimented with using hospital number alone as the metric for healthcare capacity, i.e., without adjusting for service population size. We obtained similar results and have now included it in this draft as a sensitivity analysis.

In Methods, under "Assessing the Healthcare capacity and COVID-19 related Healthcare pressure", we provide further clarification of the metric used, as well as clarifying the use of the alternative metric as a sensitivity analysis:

In mainland China, Grade II and III hospitals have 100-499 or 500+ hospital beds, respectively, and are equipped with ventilators³⁸. Thus, they are more important compared to community hospitals and clinics for COVID-19 management. Healthcare capacity in prefecture i (HC_i), therefore, can be expressed as:

$$HC_i = \frac{n_{hospital,i}}{Pop_{residential,i}}$$

where $n_{hospital,i}$ is the number of Grade II and III hospitals in prefecture i , and $Pop_{residential,i}$ is the residential population of prefecture i . We use the size of residential population from the China Statistics Yearbook²⁹. Healthcare pressure (HC_i) is stratified into high and low settings by taking the top and 50% of prefectures with available data. Note that this metric cannot accurately reflect the prefecture level population sizes during LNY. Take Beijing as an example, although the residential population sizes is approximately 22 million, over 10 million left the city for the holidays³⁸.

Healthcare pressure in prefecture i during week w ($HP_{i,w}$) was calculated by dividing weekly confirmed COVID-19 cases³¹ by the healthcare capacity:

$$HP_{i,w} = \frac{n_{confirmed,w}}{HC_i}$$

where $n_{confirmed,w}$ is the number of confirmed COVID-19 cases during week w .

We considered an alternative measure of healthcare capacity that does not adjust for the background residential population sizes:

$$HC_i = n_{hospital,i}$$

We also included a brief discussion in the main text of the Results:

This pattern persisted when we used an alternative healthcare capacity measure of the number of Grade II and III hospitals without adjusting for

background population size (Supplemental Figure 16).

And

Using the alternative healthcare capacity measure that considers number of hospitals only, we found similar relative associations with a slight delay (Supplemental Figure 16).

And included a Supplementary figure showing a similar pattern to the main text figure:

Supplemental Figure 16. Measuring healthcare capacity.

Assessing healthcare capacity related to COVID-19 using an alternate metric, the number of Grade II and Grade III hospitals in each prefecture, without adjusting for population size.

We were not able to calculate the population standardised incidence as, given the holiday travels around LNY, we cannot accurately calculate the correct dominator for incidence.

Minor changes

Reviewer 1, Minor 1

The finding that the temporary increase in outbound travel from Wuhan just prior to the cordon sanitaire was not unique to Wuhan is a valuable one. I am not sure that I fully understand the metrics used (page 14, eq 3 and eq 4), however. I think this is related to the notation. For $V1_i$, for instance, is the first term the peak daily outbound Baidu mobility index score for unit i

in the time period of interest in 2020 divided by the mean daily score for the same location in 2019 (minus 1)? In other words, this is just the relative change from mean 2019 mobility index to peak 2020: $(\text{peak_2020} - \text{mean_2019})/\text{mean_2019}$. If so, I would clarify in the notation of the equations that this is the peak 2020 value in the time window in question. This is currently unclear, as F is defined as a daily metric earlier in the methods section.	
Response	We agree with the reviewer that further clarification is needed describing how we assessed the existence of surge outflow from Wuhan preceding the cordon sanitaire. We have rewritten the methods section to more clearly represent the two equations used to calculate each variability metric. We also clarify the specific time periods under consideration.  In Methods, under “Temporal Trajectory Analysis and Surge Evaluation”: We quantified the peak outflow from each prefecture in the five-day window before LNY (i.e., two to seven days before LNY, zero to five days before the cordon sanitaire). We used two parameters to characterise the magnitude of the change in outflow in 2020 compared to 2019 in each prefecture i: $V_{1,i} = \frac{\text{mean}(F_{i,\tau_{2020}})}{\text{mean}(F_{i,\tau_{2019}})} - 1 \quad (\text{eq 4})$ $V_{2,i} = \frac{\text{mean}(F_{i,\tau_{2020}}) - \text{mean}(F_{i,\tau_{2019}})}{\text{std}(F_{i,\tau_{2019}})} \quad (\text{eq 5})$ where $F_{i,\tau}$ is the total Baidu outflow from prefecture i in the time period τ. τ in 2019 corresponds to 29 January - 3 February 2019, and in 2020 corresponds to 18 January – 23 January. The dates are different each year because they are aligned to the date of LNY in 2019 and 2020.

Reviewer 1, Minor 2	
SI Fig 13: Unless I missed it, the derivation of these distance kernel plots is not described in neither the methods nor SI. I would suggest that either this analysis be described in more detail in either the methods section or in the SI / figure caption, as it is otherwise very difficult to interpret.	
Response	We have added a section to Methods and extended the caption in the SI figure. The methods section now reads:

	Distance Kernels To determine how the relationship between distance and travel flow changed over Chunyun and in response to the cordon sanitaire, we calculated the frequency of journeys of at least distance n kilometres, for n up to the maximum distance 4185km, on each day of the study period. These plots are shown for Beijing, Guangzhou, Shanghai and Wuhan, for both inflow and outflow in Supplemental Figure 14.
--	---

Reviewer 1, Minor 3	
SI Fig 15: Community modularity. Is it possible to identify which blue node corresponds to Beijing, Shanghai, and Guangzhou/Shenzhen? This would add some useful information to these interesting plots.	
Response	We have changed the colour in both panels of the figure to better distinguish the different communities. We have also added a colour legend which helps clarify the colours in both panels of the figure.

Reviewer 1, Minor 4	
I believe that reference 6 should be Lai, S., Ruktanonchai, N.W., Zhou, L. et al. Effect of non pharmaceutical interventions to contain COVID-19 in China. Nature (2020). https://doi.org/10.1038/s41586-020-2293-x . It is possible that there is a manuscript with the same name in the journal mentioned here, but seems more likely that this is an outdated reference (the manuscript appears to have now been accepted for publication in Nature).	
Response	Thank you for pointing this out. We have amended this reference: Lai, S. et al. Effect of non-pharmaceutical interventions to contain COVID-19 in China. Nature (2020) doi:10.1038/s41586-020-2293-x.

Reviewer 1, Minor 4	
Reference 18: is "People" the correct author name?	
Response	We have amended this reference. Additionally, we have rechecked the entire reference library as we noticed our reference manager may have some problems with importing files. We therefore re-generated the entire reference list.

	Xiong, Jian. 346 medical teams of 42,000 people arrived in Hubei to fight the epidemic. People's Daily Overseas Edition http://paper.people.com.cn/rmrbhwb/html/2020-03/09/content_1975139.htm (2020).
--	--

Reviewer #2

Major changes

Reviewer 2, Major 1	
The ordering of the sections is jarring. Rearrange the content of the paper so that the methodology comes before the results section (why on earth does it not at present?). As it is, the reader is presented results without explanation and this is maddening – I've just spent half an hour trying to piece together what is being clustered in Figure 1 and it turns out this information comes later in the paper (after the discussion?!?).	
Response	The manuscript is arranged in the format required for Nature Communications manuscripts. To ensure that readers are not confused, we have edited the text so that each section leads more smoothly to the next.

Reviewer 2, Major 2	
no clear research aims are articulated in the introduction other than a vague notion that we need to understand the effectiveness of travel cordons and human mobility if we are to better understand the spread of disease in a pandemic. Yes, OK, but what do you really want to find out? Is it how specifically the cordon disrupted regular travel patterns and whether this had some kind of spatial dimension? Is it that flow patterns have interesting relationships with the size and spatial arrangement of other cities in China and if we understand how, for example, people tend to flow from big cities back to smaller towns at this time of year? Are we then able to say something about likely disease spread and propagation under different (more normal) travel regimes? Where does the hospital and network analysis fit into this picture? None of this is very clear at all and a much better job needs to be done at the beginning of the paper to set this up. Reference to the wider literature feels a bit scant at the beginning and a better review might set up a more successful articulation of research aims. More appears later on, but again, refer to my very first point.	
Response	We thank the reviewer for this comment and agree that the paper will be improved by a clearer statement of research aims at the beginning of the

paper.

We have modified the text in the introduction to more clearly state the aims of this work and explain our motivation for using the variety of techniques in this paper.

We have added the following text to the introduction:

A key unknown is to what extent the observed travel patterns in Wuhan and the rest of China were part of regular seasonal movements or were responses to the emerging epidemic or interventions against it, including the cordon sanitaire. Relying on a range of data scientific techniques, we examined human movement between Chinese prefectures on multiple geographic scales to provide a detailed examination of travel patterns during the early stages of the COVID-19 pandemic in China. We combined analyses of travel patterns from Wuhan, the centre of the first COVID-19 epidemic and the first Chinese city to introduce large scale movement restrictions, with an analysis of the effects on the overall Chinese travel network. We further explored the relationship between travel patterns during the LNY holidays and regional healthcare capacity, to understand the impact of the human movements on the healthcare pressure caused by the spreading epidemic. This research is intended to provide a complete picture of the overall movement dynamics in China, and the public health implications of those movements, and has relevance to other countries implementing travel restrictions in an effort to limit the spread of COVID-19.

We have also made an effort to better contextualize this research in the introduction, discuss existing work that has been published using the same data sources, and explain the novelty of this work compared to the existing published research.

Several studies have focused on assessing the effectiveness of the cordon sanitaire in Wuhan and other domestic travel restrictions in China in the context of COVID-19 control⁶⁻⁸. As other affected regions worldwide begin implementing similar travel restrictions⁹, it is critical to understand human mobility patterns during the initial phase of the COVID-19 pandemic and their potential implications for other countries.

And:

Previous analyses of Baidu movement data have used mobility data in transmission models^{6,11}, and others have examined the changes in patterns around Wuhan⁷.

Reviewer 2, Major 3

results are presented, but without clear linkage to specified aims or objectives. For example, a section in the results refers to movement between prefectures and cities in different population quartiles and there is reference to volumes of travel, patterns related to distance, and a very difficult to interpret graph (sup fig 14) related to mobility and population size quartiles. There's lots here, but the authors do an inadequate job of both interpreting these graphics for the reader and linking the narrative back to so pre-specified objectives.

The conclusion on p7: "Therefore, medium sized locations could play a key role in limiting the spread of COVID-19 to prefectures with fewer residents" feels a little like some straws are being clutched at - if they could, then how? It made me think this could well be an urban hierarchy thing. Read David Plane's paper on migration up and down the urban hierarchy - <https://www.pnas.org/content/102/43/15313> - movements of people have been shown to go up and down the hierarchy with larger steps between places that are closer in the hierarchy to each other than those that are further apart. I think the patterns that you are unearthing here – which are very interesting, don't get me wrong – are demonstrating this phenomenon and conclusions like this need to be a little grounded a little more in this sort of theory.

Response

We thank the reviewer for this comment and for suggesting the relevant reference. In this study, we have observed patterns of travel between prefectures that are supported by existing theory about hierarchical travel flows. Although the Baidu migration index measures daily movement, during the weeks immediately before LNY, it can be interpreted as a proxy for long-term migration due to the large percentage of holiday travel, which tends to be people returning to family homes, i.e. the homes they have migrated from. Thus, the pattern we observed immediately before LNY can be explained the urban hierarchy previously described by Plane *et al* as people temporarily return to where they migrated from for the holidays.

To address this valuable comment, we have made the following amendments to the text:

- In supplemental figure 14 (now 13), we have revised the caption to clarify why this information was not included in the main text:

Each line represents a prefecture. The blue line represents the average levels for a given direction, within a given population quartile. In- and outbound travel volumes show similar trends over time. Over the two to three weeks prior to the Lunar New Year (LNY), travel activities stabilised at relatively high levels. Then, during the week after LNY, travel activity significantly dropped. As time goes on, travel activities gradually recover. By the end of this study, travel volumes have not been able to recover to their pre-LNY/ COVID-19 levels. Despite of similar trends, the specific compositions of these travel activities are distinctive based on population size and travel directions. More discussion can be found in the main text, centered around Figure 2.

	 In Results, section on “Movement patterns cross China”, we provided the following revision: This analysis of origin or destination locations revealed diverging hierarchical effects, rather than a simple cascading flow of travelers from larger to smaller population prefectures. Travellers from large prefectures more often travelled to other large or medium size prefectures; travellers from medium and small prefectures more often travelled between medium and small prefectures. Holiday travel immediately preceding LNY can be considered an indicator of long-term migration in China, as people travel back along their long-term migration route temporarily to visit family. The patterns we observed are consistent with the migration “step” effect along the urban hierarchy, in which geographic regions of similar population size exchange members more often^{15,16}. The divergence in hierarchical flow between high and low population prefectures means that middle population prefectures could play a key role in limiting the spread of COVID-19 to prefectures with fewer residents. Non-pharmaceutical interventions could target these medium-sized prefectures to prevent epidemics from reaching the relatively rural parts of China. We provided an additional reference to elaborate on the “step effect” in the context of China – although larger jumps between farther urban hierarchy was observed during the Mao period (as it was encouraged/mandated by government policy), contemporary migration patterns do show smaller jumps along the urban hierarchy: 16. Hao, P. & Tang, S. Migration destinations in the urban hierarchy in China: Evidence from Jiangsu. Population, Space and Place 24, e2083 (2018).
--	---

Reviewer 2, Major 4	
having read through the paper, I am still none the wiser about what you have actually clustered. Talk of trajectories is confusing as I was immediately thinking of directional trajectories and I could find no direction. I think, now, after going over the paper a couple of times, that the trajectories are temporal ones, but I am still not 100% sure. This could all be cleared up very easily with a section that defines the variables more clearly with a better notation (see some of my suggestions below).	
Response	We have clarified this at the first mention, and we have also amended the text in the caption of Figure 1. The main text now reads:

	Using k-means clustering of the timeseries of daily outbound travel from Wuhan to other prefectures, we identified four general temporal clusters that captured the travel patterns from Wuhan (Figure 1e). The caption of Figure 1 now reads: The clusters are defined by k-means clustering of the timeseries of outbound travel volume (see Methods). The timeseries have been normalised by the total flow of each, to allow comparison of the profile.
--	--

Reviewer 2, Major 5	
it feels a bit of a hodgepodge of techniques and methods – which are all pretty cool, I’m sure, but none feel like they are given the attention they deserve in the context of a well-thought-out research narrative. The access to health care bit – yes, very interesting, but it’s just stuck in there and doesn’t really seem to fit in with the migration stuff. I think the migration bit, the network analysis bit and the hospitals bit could probably all be separated out into different papers and if given the due care and attention required – all make very interesting papers in their own right – as I’ve already said, there’s some good stuff in here. I feel a bit like Greg Wallace on Masterchef feeling a bit sad when the contestant throws too many great ingredients into the dish and it sort of doesn’t work.	
Response	We have edited the introduction to more clearly delineate where this manuscript closes gaps in the existing literature. We use a range of data scientific techniques which allows us to examine different dimensions of the movement patterns, as well as the implications of these movements. We agree that further analyses could be made of many aspects of the movement patterns in China in early 2020, but those lie outside of the scope of this study. In response to the reviewers comment we have edited the text to highlight better the aims, and the Discussion to highlight the need for further work on this topic.  • We extended the introduction paragraph to highlight better the range of data scientific techniques used, as well as the gap in the literature: Previous analyses of Baidu movement data have used mobility data in transmission models^{6,11}, and others have examined the changes in patterns around Wuhan⁷. A key unknown is to what extent the observed travel patterns in Wuhan and the rest of China were part of regular seasonal movements or were responses to the emerging epidemic or interventions against it, including the cordon sanitaire. Relying on a range of data scientific techniques, we examined human movement between Chinese prefectures on multiple geographic scales to provide a detailed examination of travel patterns during the early stages of the COVID-19

	pandemic in China. We combined analyses of travel patterns from Wuhan, the centre of the first COVID-19 epidemic and the first Chinese city to introduce large scale movement restrictions, with an analysis of the effects on the overall Chinese travel network. We further explored the relationship between travel patterns during the LNY holidays and regional healthcare capacity, to understand the impact of the human movements on the healthcare pressure caused by the spreading epidemic. This research is intended to provide a complete picture of the overall movement dynamics in China, and the public health implications of those movements, and has relevance to other countries implementing travel restrictions in an effort to limit the spread of COVID-19.  • We have added the following to the Discussion: This study analysed the human mobility patterns around China during different stages of the local COVID-19 epidemics, from early Chunyun to Wuhan’s cordon sanitaire and other travel restrictions. Using a range of techniques, we assessed the patterns of movement specific to Wuhan and the characteristics of the travel network throughout China considering the implications of changing travel patterns on the spread of COVID-19. We also explored the impact of travel patterns on Chinese prefectures, assessing the changes in healthcare pressure due to varying patterns of human mobility typically associated with LNY, which coincided with the early stages of the COVID-19 pandemic. Many countries have now implemented similar travel restrictions to reduce disease transmission. Understanding the implications of travel patterns before, during, and following travel restrictions is valuable for informing public health interventions, surveillance, and healthcare demand planning globally.
--	---

Minor changes

Reviewer 2, Minor 1	
P2 – line 13 – ‘largest annual human migration’ – is this migration or merely seasonal visitation? How long to people spend at their new location after migrating? Week, 2 weeks? Longer? Would also help to clarify the main purpose of this movement – is it to visit family back home? Are these workers? Students? Both? Similar to the Christmas movements in the UK or Thanks Giving in the US? Or are there alternative cultural, spiritual or other motivations? Yes, it might be mentioned when following the link, but a quick note would help very much here.	
Response	We thank the reviewer for this comment about Chunyun . This is indeed

important, and puts the rest of our discussion into context.

Chunyun is seasonal travel for the Lunar New Year (LNY). Depending on occupation, people spend different amount of time in their holiday locations. Migrant workers (primarily in manufacturing and construction) and university faculty/students tend to stay in their holiday locations for up to a month's time; private sectors and foreign companies tend to give their employees up to two weeks off; public sector employees tend to only get a week off. Migrant workers and university students also tend to travel a bit earlier because their flexibility in schedule. In a way, it is similar to Thanksgiving in the US and Christmas in the UK. However, it is important to note the scope of this travel surge due to the holidays. Before the LNY of 2020, ten million people left Beijing, the capital of China with a population of 22 million.

We also appreciate the question about the terminology. We used the term "migration" in this study to describe the travel flows between prefectures simply because it is the literal translation of the metric that Baidu Huiyan uses to describe their original data, i.e., the Migration Index. We fully acknowledge that this terminology may imply something longer term to some readers. Thus, we have changed the terminology from "migration" to "movement" where we can. When kept the "migration" in the phrase "migration index" so readers can make direct references to what information we are using from Baidu Huiyan.

To address this comment from the reviewer, we made the following changes:

- Throughout the manuscript, we changed the term "migration" to "movement". The changes can be seen in the tracked changed versions of this manuscript. We will not reflect on each one of these changes in the response letter.
- In Introduction, we added the following description:

The purpose of this holiday travel is often to visit family members. The temporary displacement from residential addresses as a result of this holiday travel could last one to two weeks, up to a month.

- In Methods, we added the following description to describe the specific scope of *Chunyun*:

Note that this metric cannot accurately reflect the prefecture level population sizes during LNY. For example, the residential population size of Beijing is approximately 22 million, and over 10 million left the city for LNY⁴¹.

Reviewer 2, Minor 2

P3 – how are you determining out-bound travel? Is there some sort of distance threshold or change of region or something that you are using to count out-migrants? Need some clarity here.

Response	Outbound and inbound travel is determined in the Baidu migration index. We have clarified the definition of outbound travel: In the methods section we have added the following explanation: Baidu movement flow index is collected in 8-hour windows and is provided as origin-destination flows between pairs of prefectures. We further processed this data to produce symmetrical matrices of daily travel between all Chinese prefectures.
----------	---

Reviewer 2, Minor 3

P4 – when you refer to a ‘trajectory’ (the object being clustered) what does this mean? Trajectory would normally imply some kind of direction and distance dimension, but is it not clear how these are incorporated into the objects being clustered

Response	Thank you for this question. We used the term for a directional timeseries (in- or outbound). However, to avoid confusion we have clarified the main text in Results and in the caption of Figure 1: • The Results, section “Human movement surrounding the epicentre – Wuhan, Hubei”, we provide the following revision:Using k-means clustering of the timeseries of daily outbound travel from Wuhan to other prefectures, we identified four general temporal clusters that captured the travel patterns from Wuhan (Figure 1e).• In the revised caption of Figure 1e, we have included the following explanation:e), outbound travel trends from Wuhan to the most connected prefectures in China, stratified by clusters with similar temporal trajectories. The clusters are defined by k-means clustering of the timeseries of outbound travel volume (see Methods). The timeseries have been normalised by the total flow of each, to allow comparison of the profile.
----------	---

Reviewer 2, Minor 4

P4 – how are you defining ‘neighbours’? Are these all other prefectures in China that are not Wuhan? Or just those we would normally consider as ‘neighbours’? e.g. contiguous zones or those within a certain distance. It would be hard to consider Beijing a neighbour, for example. If not neighbours, then ‘other prefectures’ is fine, but the distinction needs to be clarified.	
Response	Thank you for pointing this out. By neighbours, we meant all other prefectures with movement from Wuhan. We have removed “neighbours” and have replaced it with the following title for Figure 1: Travel patterns between Wuhan and other connected prefectures.

Reviewer 2, Minor 5	
P4 – I am struggling to see much difference between cluster A and B from figure 1 – how is the cluster defined? This goes back to the earlier point about clearly defining the objects that are being clustered here. Supplementary material – the values in the matrices are what? The values appear to correspond to numbers in the supplied data matrices on github, but the y axis on each graph is not labelled.	
Response	Clusters of timeseries trajectories of outbound travel from Wuhan (and other cities included in the appendix) are determined by k-means clustering. Due to the differences in terms of travel flow magnitude, we rescale these timeseries trajectories to make them more comparable. The k-means clustering will group similarly shaped timeseries trajectories together into the same cluster. In this case, B has a higher early-January peak and A has a more pronounced pre-LNY peak. We performed sensitivity analysis on the number of clusters, examining the effect of different numbers of clusters on the resulting clustered trajectories, described in the methods We have expanded the methods section to describe the original format of the data and provide further details on the processing steps we used to produce the symmetrical matrices available on GitHub.  The methods now reads: In order to characterise the shape of the outflow trajectories rather than the magnitude of certain outflows, we calculated the normalised flow, N, between origin (i) and destination (j) prefectures on each day (t), by dividing the outflow measured by the travel index T, by the total movement between the 1st and 23rd January 2020, as: $N_{i,j,t} = \frac{T_{i,j,t}}{\sum_{t=1}^{t=23} T_{i,j,t}} \quad (\text{eq.3})$

Reviewer 2, Minor 6

OK, you are making me work hard here and I'm piecing together the information slowly – Fig 1 now refers to a 'normalised migration index' and the legend explains that this is 'normalised by the total of each flow'. But I have no idea whether this is total outflows, total inflows or both? In notational form is M is the migration flow between origin i and destination j, the normalised flow could effectively be either $M_{ij}/(\sum_j M_{ij})$ or $M_{ij}/(\sum_i M_{ij})$ or $M_{ij}/(\sum_i \sum_j M_{ij})$ but this only assumes a single temporal dimension.

Response

The trajectories are the outbound timeseries from Wuhan to other prefectures. Normalisation is by total flow between each pair of prefectures over the time period of the study. This allows us to study the shape, rather than the magnitude of each trajectory.

We have expanded the description of the data and the use of inbound and outbound time series (Section "Mobility Data"). We have also clarified how trajectories are normalised and that the trajectories being clustered are outflow trajectories (Section "Temporal Trajectory Analysis and Surge Evaluation"). We have also added a new equation (eq. 3), which specifically describes the method for normalising outflow trajectories.:

In order to characterise the shape of the outflow trajectories rather than the magnitude of certain outflows, we calculated the normalised flow, N , between origin (i) and destination (j) prefectures on each day (t), by dividing the outflow measured by the travel index T , by the total movement between the 1st and 23rd January 2020, as:

$$N_{ij,t} = \frac{T_{ij,t}}{\sum_{t=1}^{23} T_{ij,t}} \quad (\text{eq.3})$$

Reviewer 2, Minor 7

Right, I have read on a little further and the methods section, bizarrely, comes after the results section. This is very odd – please rearrange this so that your future readers do not have the same period of head scratching as I did!

Response

The manuscript ordering is in Nature Communications required structure.

Reviewer 2, Minor 8

P4 – “Compared to Cluster A, Cluster B and C detected their first COVID-19 cases at approximately the same time (Fig 1d). Prefectures in cluster D confirmed their first cases 1.08 days earlier (Figure 1d).” – I think you mean 1b?

Response Thank you for pointing out this glitch. We have amended the figure number.

Reviewer 2, Minor 9

P7 – Healthcare availability and migration. Interesting on one level and it kind of makes sense in the context of larger cities having better hospital facilities and most LNY movements moving down the urban hierarchy (one assumes students and workers going back to villages and smaller towns to visit family). This is presented without much further comment, however. Did this lead to an excess of deaths or was it significant in any other way, or do we simply not know? It feels like a bit of a bolt on and again, suffers from the lack of clear research objectives.

Response Thank you for this comment. We are able to demonstrate the association between healthcare capacity and the COVID-19-related healthcare pressure, and illustrate that COVID-19 related burden was not distributed proportionally, and in this case low healthcare capacity prefectures were facing higher COVID-19-related healthcare pressure. We have elaborated on the importance of this result:

The movement observed was associated with COVID-19-related healthcare pressure (see Methods), a measure of confirmed cases compared with healthcare capacity (Figure 3b). From the week before LNY to two weeks after, locations with low healthcare capacity experienced significantly higher pressure compared to locations with high healthcare capacity. Therefore Chunyun not only increased the chance of infection along mobility networks, but also shifted healthcare pressure caused by COVID-19 to regions with low healthcare capacity¹⁷⁻¹⁹. Using the alternative healthcare capacity measure that considers number of hospitals only, we found similar relative associations with a slight delay (Supplemental Figure 16).

In the introduction, we have clarified the research objectives:

We further explored the relationship between travel patterns during the LNY holidays and regional healthcare capacity, to understand the impact of the human movements on the healthcare pressure caused by the spreading epidemic.

And in results we revised the section header to more accurately reflect the goal and the content:

	Healthcare capacity and COVID-19-related Healthcare Pressure
--	---

Reviewer 2, Minor 10	
P8/9 and Fig 4, reference to ‘modularity’ but no definition – yes a link is given, but again, it would help the reader if this were briefly explained. I think there is something interesting going on with the lead up to LNY and the Wuhan cordon in the community patterns, but again, I am not sure what I am to be taking away from this.	
Response	 • We have added the following sentence to the main text: Each community (or module) has more connections within vs between communities, and modularity is one method for measuring community structure in networks. • And we have added the following to the caption of figure 4: Community structure is measured through “modularity”, a metric defining the strength of connections within vs between communities. The members of communities are determined by the Leiden algorithm²⁰

Reviewer 2, Minor 11	
P10 – “We found a limited relationship between spatial proximity and epidemic spread where larger, distant populations detected their first COVID-19 cases earlier than smaller locations that are closer to Wuhan.” – this is interesting and suggests to me that the standard gravity models of migration are probably holding quite well here. Given that the authors are in possession of all of the ingredients to fit a gravity model of migration (population data, origin/destination migration matrices, distance matrices), I would recommend that if this were carried out, they would be able to comment more effectively on the deviations from expected flows. This could be done in using a simple Poisson or negative binomial regression model – in R, something like: <code>migration_model <- glm(Tij_flow ~ log(origin_pop)+log(destination_pop)+log(dist_ij), family = poisson(link = "log"), data = some_pairwise_rearrangement_of_china_prf_connectivity_0101_0301_plus_the_orig_dest_and_distance_variables)</code> – it would then be possible to compare fitted values with observed flows and comment on whether the spread of the virus from Wuhan actually did something that we would expect, given the things we know normally influence migration flows, or did it do something quite different?	
Response	Thank you for this comment. We agree that the overall patterns of movement – large, distant prefectures are connected more than small prefectures – may

follow the general rules of the gravity model.

Unfortunately, we do not have the measures needed to fit a model like this. For population sizes, we are using the resident population sizes from the China statistics yearbook 2018. This is an annual average of a value that fluctuates quite dramatically, as it accounts for China's large floating population ("migrants who moved between provinces or counties and resided at their destinations for six months or more") [<https://doi.org/10.1111/j.1728-4457.2004.00024.x>].

Our study period includes the Gregorian calendar New Year (which is also a national holiday in China), *Chunyun* (the 40-day travel surge before and after the Lunar New Year), Lunar New Year, and various COVID-19 related movement restrictions. None of these periods likely resemble the annual average travel patterns and we cannot accurately quantify the deviation from population values recorded in the China statistics yearbook. Additionally, we cannot use the migration index to characterise the change in a city's resident population because Baidu Huiyan does not provide the algorithms needed to convert migration index to the actual number of travellers (likely for privacy reasons). Thus, unfortunately, important ingredients of gravity models are missing.

Because of this issue with the population data and the uniqueness of the time periods we are studying, we have only been able to use the resident population in this study to calculate healthcare capacity, where the resident population size captures the annual average population size that relies on the healthcare infrastructure in a given prefecture.

Additionally, we feel that models are most useful when data are not available, and in this case, there is a wealth of available data on the movements of people between prefectures. We, therefore, feel that fitting a model which would approximate the flow from one place to another would not contribute to the analysis. For researchers who may wish to analyse movements outside the time period where data are available, a gravity model or other frameworks could be investigated, but we feel it is outside of the scope of this data-driven exploration of observed patterns at the early stage of the pandemic in China.

To clarify these points:

- In Methods, under section "Demographic and Healthcare System Data", we provide the following revision:

The 2018 resident population sizes were retrieved from the China Statistics Yearbook³¹. This metric accounts for migrant populations, and thus is expected to fluctuate during holiday seasons.

- In Methods, under section "Assessing the Healthcare Capacity and COVID-19 related Healthcare Pressure", we provided the following context:

	For example, the residential population size of Beijing is approximately 22 million, and over 10 million left the city for LNY⁴¹
--	---

Reviewer 2, Minor 12

P13 – There’s some neat data wrangling here, but it could all be specified a little more clearly. I’m not sure equation 2 is quite correct. The ‘overall migration index’

The volume of migration, T_{ij} , between origin, i and destination, j can be calculated:

$$T_{ij} = O_i * P_{ij,O} = D_j * P_{ij,D}$$

Where the overall Baidu Migration Index outbound from origin, O_i and inbound to destination, D_j for $N=366$ prefectures:

$$O_i = (\sum_j T_{ij}) / T$$

$$D_j = (\sum_i T_{ij}) / T$$

And two alternative sets of probabilities can be generated:

$$P_{ij,O} = T_{ij} / O_i$$

$$P_{ij,D} = T_{ij} / D_j$$

It is assumed that any time dimension, t , is implicit here with each of these calculated for each of the $t = 1 \dots 61$ days in the dataset.

Response	We thank the reviewer for catching this. In methods, under “Mobility Data”, we provided the following revision: We calculate the volume of human mobility between each pair of prefectures on each day between 1st January 2020 and 1st March 2020 using the following equation: $T_{ij,t} = F_{i,outbound,t} * p_{ij,outbound,t} \text{ (eq.1)}$ Where $T_{ij,t}$ is the volume of mobility from location i to location j on day t, F is the overall Baidu migration index with direction (inbound or outbound) at location i, and $p_{ij,outbound}$ is the proportion of all outbound travel that originated in i and ended in j. References to “inbound” and “outbound” travel are made in regards to a specific origin or destination location. We further
----------	--

	validated this measure by assuming that inbound and outbound were equal, as: $T_{ij,t} = F_{i,outbound,t} * p_{ij,outbound,t} = F_{j,inbound,t} * p_{ij,inbound,t} \text{ (eq.2)}$ where $p_{ij,inbound,t}$ is the proportion of all inbound travel that end in j and originate in i. Note that $p_{ij,outbound,t}$ and $p_{ij,inbound,t}$ are only available for the top 100 connected prefectures. In other words, $p_{ij,outbound,t}$ is only available for the top 100 destinations originating in i; $p_{ij,inbound,t}$ is only available for the 100 origins with the most travellers to j. We were not able to validate for $T_{ij,t}$ in the cases where $p_{ij,outbound,t}$ and $p_{ij,inbound,t}$ are not simultaneously available. Using data from Baidu Huiyan, we created a symmetric, 366*366 connectivity matrix for each day between 1 January 2020 and 1 Mar 2020 (61 days).
--	---

Reviewer 2, Minor 13	
P13/14. Discussion of trajectories – I am still struggling to understand what this trajectory is? Please explain it with reference to the sort of notation I describe above. I think a trajectory is a T_{ij} flow? But then there is talk of an ‘outflow trajectory’ which can only be O_i as described above? However, I could contest that this is a trajectory as it has no directional component to it (as implied by the term). Again, just not clear and it needs clarifying. Although I am thinking as I go here, and now I am thinking that the trajectory has a temporal rather than directional dimension. This makes more sense, but I should not have to be guessing this.	
Response	A trajectory is a time series of daily flow from one prefecture to another, so it has both time and direction. To avoid any confusion for readers, we have changed the nomenclature to timeseries, and we have added a definition where first used. The definition reads: Using k-means clustering of the timeseries of daily outbound travel from Wuhan to other prefectures, we identified four general temporal clusters that captured the travel patterns from Wuhan (Figure 1e).

Reviewer 2, Minor 14	
P14 – following the notes above, please rethink the notation in eq.3 & 4 to incorporate time more effectively.	
Response	We have added the t subscript to make these equations clearer. They now read:

We calculate the volume of human mobility between each pair of prefectures on each day between 1st January 2020 and 1st March 2020 using the following equation:

$$T_{ij,t} = F_{i,outbound,t} * p_{ij,outbound,t} \text{ (eq.1)}$$

Where $T_{ij,t}$ is the volume of mobility from location i to location j on day t , F is the overall Baidu migration index with direction (inbound or outbound) at location i , and $p_{ij,outbound}$ is the proportion of all outbound travel that originated in i and ended in j . References to “inbound” and “outbound” travel are made in regards to a specific origin or destination location. We further validated this measure by assuming that inbound and outbound were equal, as:

$$T_{ij,t} = F_{i,outbound,t} * p_{ij,outbound,t} = F_{j,inbound,t} * p_{ij,inbound,t} \text{ (eq.2)}$$

where $p_{ij,inbound,t}$ is the proportion of all inbound travel that end in j and originate in i . Note that $p_{ij,outbound,t}$ and $p_{ij,inbound,t}$ are only available for the top 100 connected prefectures. In other words, $p_{ij,outbound,t}$ is only available for the top 100 destinations originating in i ; $p_{ij,inbound,t}$ is only available for the 100 origins with the most travellers to j . We were not able to validate for $T_{ij,t}$ in the cases where $p_{ij,outbound,t}$ and $p_{ij,inbound,t}$ are not simultaneously available. Using data from Baidu Huiyan, we created a symmetric, 366*366 connectivity matrix for each day between 1 January 2020 and 1 Mar 2020 (61 days).

And in the section on normalisation of the timeseries we have added the t subscript as follows:

In order to characterise the shape of the outflow trajectories rather than the magnitude of certain outflows, we calculated the normalised flow, N , between origin (i) and destination (j) prefectures on each day (t), by dividing the outflow measured by the travel index T , by the total movement between the 1st and 23rd January 2020, as:

$$N_{ij,t} = \frac{T_{ij,t}}{\sum_{t=1}^{23} T_{ij,t}} \text{ (eq.3)}$$

Reviewer 2, Minor 15

P15 I'm not sure what the Mann-Whitney U test is telling me? What am I supposed to take away from the list of p-values at the top of p15?

Response Thank you for this comment. The null hypothesis of the Mann-Whitney U test is healthcare pressure in low healthcare capacity prefectures and that in high healthcare capacity prefectures are comparable. We chose Mann-Whitney U

because the data are non-normal. We conducted this test on a weekly basis and found p -values smaller than 0.05 for three weeks from Jan 22. These results have been captured by color intensity in panel B of figure 3. A significant test tells the reader that low-healthcare capacity prefectures experienced higher healthcare pressure than high-healthcare capacity prefectures in that week.

We have made the following amendments:

- In Methods section “Assessing the healthcare capacity and COVID-19 related healthcare pressure, we removed the p -values. We added the following to elaborate on the hypotheses testing:

The null hypothesis is that the healthcare pressures in low healthcare capacity settings are comparable to that in high healthcare capacity settings; the alternative hypothesis is that healthcare pressures in low healthcare capacity settings are higher than those in high healthcare capacity settings.

- In the caption of Figure 3, we provide the following revision:

Darker shade represents weeks when low healthcare capacity settings experienced significantly higher pressure than high healthcare capacity settings; lighter shade represents when differences are not statistically significant based on Mann-Whitney U test (5% type I error rate). The comparison for week 7 has p -value = 0.06.